# Recent Advancements in the Treatment of Petroleum Refinery Wastewater

**Muhammad Shettima Lawan [1], Rajeev Kumar [1], Jamshaid Rashid [2,3] and Mohamed Abou El-Fetouh Barakat [1,4,*]**

1 Department of Environment, Faculty of Environmental Sciences, King Abdulaziz University, Jeddah 21589, Saudi Arabia
2 Department of Environmental Sciences, Faculty of Biological Sciences, Quaid-I-Azam University, Islamabad 45320, Pakistan
3 BNU-HKUST Laboratory for Green Innovation, Advanced Institute of Natural Sciences, Beijing Normal University at Zhuhai, Zhuhai 519087, China
4 Central Metallurgical R & D Institute, Cairo 11421, Egypt
* Correspondence: mabarakat@gmail.com; Tel.: +966-544910070

**Abstract:** The treatment of petroleum refinery wastewater (PRWW) is of great interest in industrial wastewater management. This wastewater contains a diverse concentration of contaminants such as oil and grease, petroleum hydrocarbons, phenols, ammonia, and sulfides, as well as other organic and inorganic composites. Refinery wastewater treatment has been attempted through various processes, including physical, biological, chemical, and hybrid methods, which combine two or more techniques. This review aims to summarize current research studies involved in the treatment of petroleum refinery wastewater using conventional, advanced, and integrated treatment techniques. Furthermore, it critically highlights the efficiencies and major limitations of each technique and the prospects for improvements. Several conventional treatment techniques (basically, the physicochemical and biological processes) are discussed. In this context, advanced oxidation processes (AoPs), especially electrochemical oxidation and photocatalysis, as well as integrated/hybrid processes are found to be effective in removing the recalcitrant fraction of organic pollutants through their various inherent mechanisms. These techniques could effectively remove COD and phenol concentrations with an average removal efficiency exceeding 90%. Hence, the review also presents an elaborate discussion of the photocatalytic process as one of the advanced techniques and highlights some basic concepts to optimize the degradation efficiency of photocatalysts. Finally, a brief recommendation for research prospects is also presented.

**Keywords:** wastewater; petroleum refinery; treatment process; physiochemical; advanced; photocatalysis

## 1. Introduction

Water is one of the most valuable natural resources in the world and alongside air and soil, supports our environmental ecosystem. It is a vital resource for a variety of human activities from domestic to industrial applications and provides the living environment for marine biodiversity. Despite this integral support, global industrialization and technological advancement are associated with generating different types of wastewater released into our environmental ecosystems [1]. Besides that, the demand for petroleum resources has continued to rise in many parts of the world to enhance the economy. However, the petroleum production and refining process is also associated with the generation of large volumes of wastewater which are highly toxic even at low concentrations [2,3]. The impact of environmental pollution is widely manifested in different areas including affecting aquatic life, destroying natural land for agricultural production, and contaminating groundwater resources [4]. On the other hand, the treatment of this wastewater is also becoming a growing challenge in the petroleum industry due to its complex and dynamic nature.

Hence, different treatment techniques including physical, chemical, biological, or hybrid processes have been employed and reported in the literature. However, many of these processes have their distinct advantages and disadvantages in terms of efficiency, energy requirements and treatment costs. Considering this, more research is necessary to explore the most appropriate treatment techniques that are cost-effective and environmentally friendly. Therefore, this review article aims to critically provide a fundamental review of the existing knowledge on the conventional, advanced, and integrated or hybrid treatment techniques of PRWW and highlight some of the basic challenges or limitations of each technique as well as a discussion on prospects.

The methodology of the review is based on a search of the findings from recent studies on the conventional and advanced as well the integrated techniques which have been applied to degrade the different pollutants from PRWW. However, a few articles published from 2012 to 2015 were also considered as they presented crucial data. The major source of these articles was from the Scopus database obtained using search keywords such as "petroleum," "refinery", "wastewater pollutants", "advanced techniques", "integrated", "hybrid" and "review". To determine the range of the available research evidence and identify the literature gap on this theme, a search of a database of published literature reviews was conducted over 10 years, between May 2013–May 2023 and returned approximately 1153 papers. About 259 review papers were published in 2022 alone and 111 reviews were published in the current year 2023 (Figure 1). Furthermore, the number of published research articles on PRWW based on different treatment techniques is presented in Figure 2. The search to find the most effective treatment method for wastewater has derived significant interest, leading to more publications within these years. There have been several comprehensive reviews such as [4–10]. However, most of these reviews do not provide a combined critical report on the recent conventional, advanced, as well as integrated techniques.

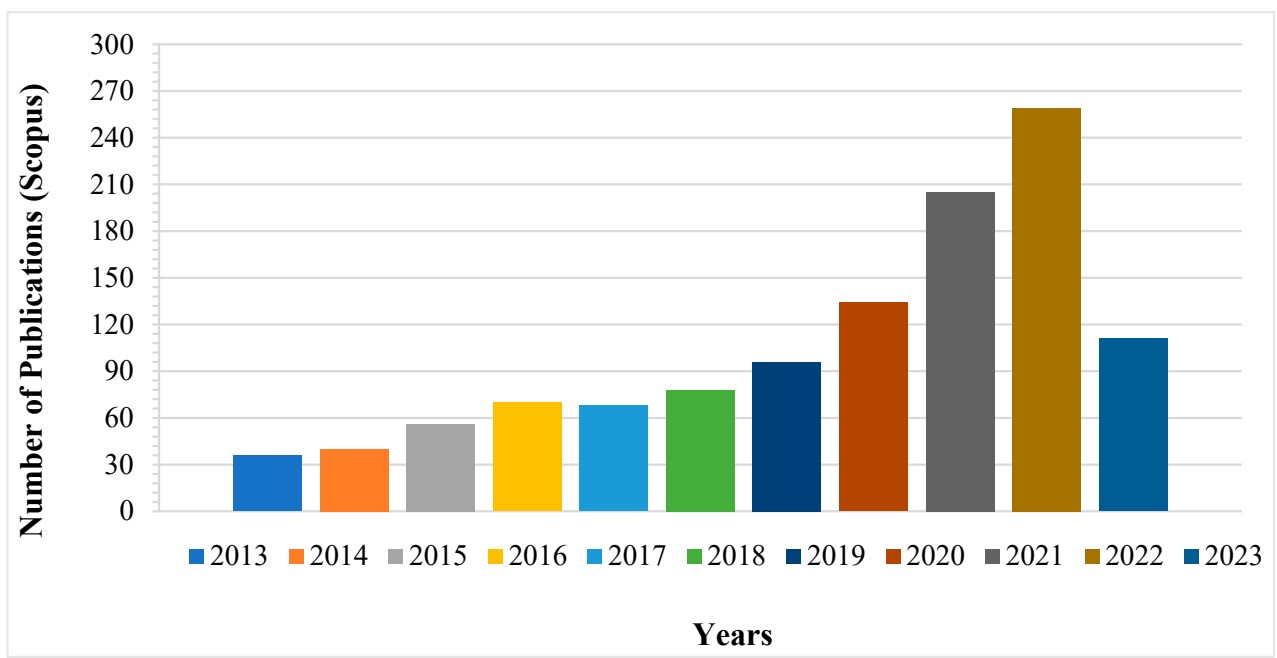

**Figure 1.** Number of published review papers on the treatment of petroleum wastewater in the Scopus database from 2013–2023.

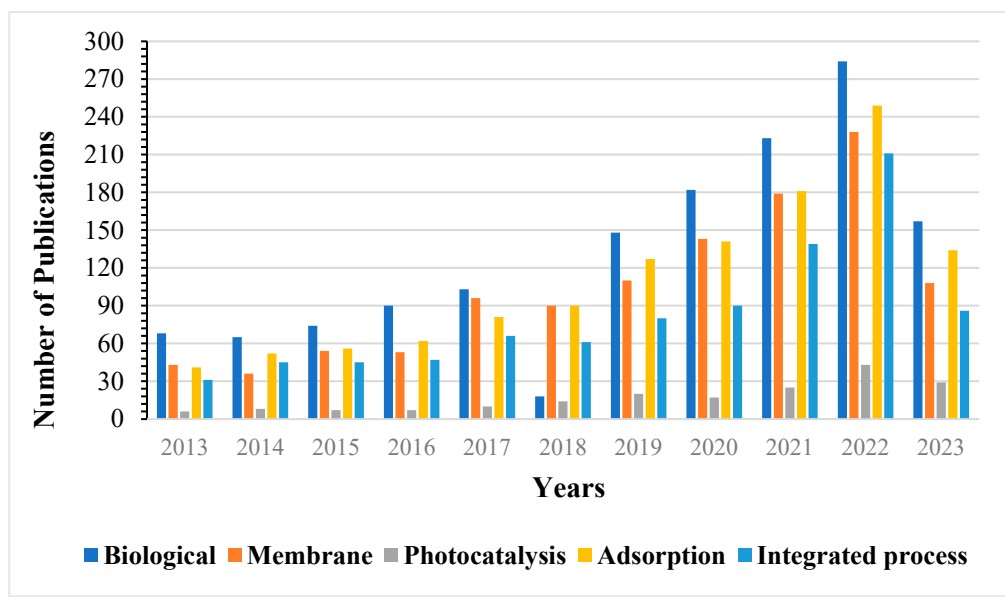

**Figure 2.** Number of research articles on the treatment of petroleum wastewater based on different treatment techniques in the Scopus database from 2013–2023.

Meanwhile, the term petroleum is used for the unprocessed oil that comes out of the ground source rock during the drilling process and is also called crude oil. It is a fossil fuel that is naturally made from the decay of plants and animals millions of years ago [11]. Petroleum refineries are complex industrial systems which are designed to refine crude oil after an exploration of various desired products through various processes (Figure 3). This categorizes the nature of the refining process into three basic stages which are: separation, conversion, and chemical treatment processes [12]. A large amount of water is required for the various refining processes which consequently generates a large amount of wastewater [10]. Meanwhile, the specific industrial operations of every refinery depend on the crude oil type and the choice of refined products. For this basic reason, almost all petroleum oil refineries are unique in their operations and hence, distinctive from one another [13].

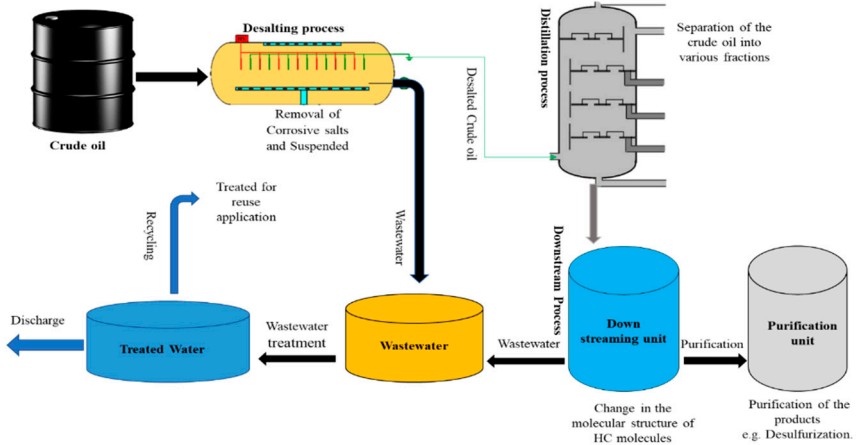

**Figure 3.** Simplified crude oil refining process and generation of the PRWW.

Ezugbe [14] reported that for every single barrel of crude oil processed, approximately ten barrels of petroleum wastewater are generated. Furthermore, literature data has indicated that about 1.6 times the volume of refined crude oil is generated as wastewater [10,15,16]. The current global output for PRWW is about 33.5 Mbps from the existing 85 Mbps of crude oil production. This global output is expected to increase by about 32%

by 2030 [10]. To be precise, a minimum of about 60–90 gallons of water (approximately 246–341 L) is reported to be used to process one barrel of crude oil [14]. The reported data indicate the huge amount of PRWW effluents continually being produced and discharged into the world's main water bodies.

Besides the generation of PRWW containing the different compositions of contaminants (Figure 4b), the operations of petroleum refineries usually have the potential to contribute to the contamination of our environmental ecosystem which also affects the land and air quality (Figure 4a). The release of toxic air pollutants as well as sludge disposal in landfills are other major threads of environmental pollution [17,18]. Similarly, the general marine biodiversity can also be affected by thermal pollution because of the disposal of hot wastewater effluent from cooling operations which can increase the temperature of the receiving body of water. Lattanzio [19], also reported that PRWW contaminants can pollute groundwater aquifers when some petroleum refining industries adopt deep well injection practices for disposal.

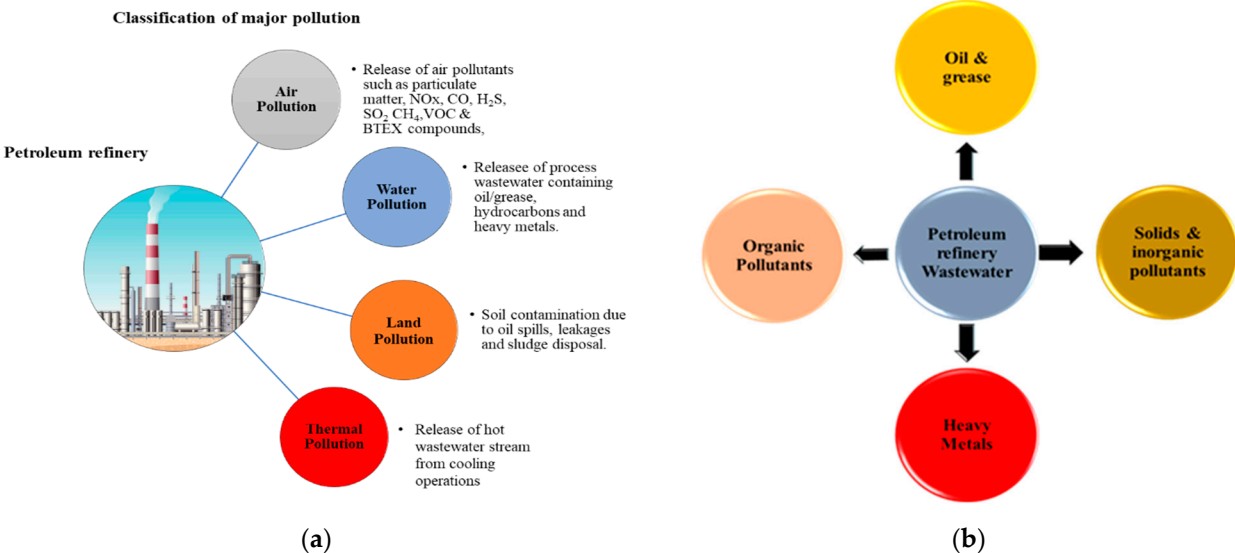

**Figure 4.** (**a**) Major environmental pollution from the petroleum refining industry. (**b**) Major classification of PRWW contaminants.

## 2. Characterization of Petroleum Refinery Wastewater

The composition of a typical PRWW usually depends on the crude oil qualities as well as the complexity and process configuration of the petroleum refining industry. Petroleum crude oil is a complex mixture of hydrocarbon compounds of different carbon chains and other toxic organics such as phenols, which usually constitute the major organic pollutants in PRWW [3]. Generally, the PRWW contaminants (Figure 4b) can be classified into (a) oil and grease, (b) organic pollutants (which includes the hydrocarbons, organic compounds and all other biological oxygen demand (BOD) contaminants), (c) inorganic pollutants (including ammonia, nitrogen, phosphorus, chlorides and other inorganic salts), and (d) heavy metals [7]. According to Diya'uddeen et al. [20] and Whale et al. [21], the most significant contaminants that are of environmental concern include oil, phenols, suspended solids, metals, ammonia, dissolved minerals, and substances which are responsible for oxygen level depletion as a measure of (BOD) and chemical oxygen demand (COD). Therefore, PRWW treatment facilities are usually designed to be capable of the removal of both organic and inorganic contaminants. A typical PRWW effluent is usually characterized by high BOD and COD because of the overall distribution of the aliphatic and aromatic hydrocarbons, grease and emulsified oils, ammonia, cyanides and other inorganic substances from the crude oil composition. It is usually rich in hydrocarbons from the three main classes which include: (a) the paraffin comprising low-chain carbon atoms such as methane (CH4),

ethane (C2H6) and propane (C3H8); (b) the naphthenes such as dimethyl cyclopentane and cyclohexane; and (c) the aromatic compounds comprising of the benzene compounds and its derivatives [22]. The aromatics are unsaturated hydrocarbons containing at least one or more benzene rings and their derivatives, generally referred to as BTEX (Benzene, Toluene, Ethyl Benzene and Xylene) which are mostly toxic to both human and aquatic species [22,23]. In most typical PRWW samples, the average reported values for BOD and COD can be up to 400 mg/L and 600 mg/L, respectively [3,12,24]. Rahi et al. [10] also reported about 150–250 mg/L, 300–600 mg/L, and 20–200 mg/L for BOD, COD and oil concentrations in desalted PRWW effluent, respectively. They further revealed that the oil concentration can reach up to 5000 mg/L in the effluents from the bottom of tanks with a concentration of about 1–100 mg/L of benzene. Similarly, Elmobarak et al. [5] reported a COD concentration of 1200 mg/L in their review of the treatment of PRWW. However, El-Naas et al. [25], have reported a COD value in the range of 3600–5300 mg/L. A review summary of the characterization of a typical PRWW effluent presented in Table 1 from different regions of the world shows that different concentrations of COD were reported, with the lowest being 112 mg/L from Brazil and up to 74,800 mg/L from Doha, Qatar. Similarly, the concentrations of heavy metals from PRWW presented in Table 2 also vary significantly. Hence, these variations in the effluent qualities prove the dynamic complexity of the PRWW. Because of the highly complex and dynamic nature of the PRWW pollutants, it is usually difficult to understand their complete chemistry and link their toxicity to the receiving environmental ecosystem [26]. As a result, an effective treatment approach must be developed to meet the discharge regulatory requirements and for recycling purposes.

Due to their high environmental impact, discharge limits for the concentrations of various parameters are always established in every petroleum refining industry for policy compliance [27]. Different review and research articles on PRWW including [20,28–31] have reported the maximum concentration level (MCL) of the PRWW contaminants. Based on the reviews, the average effluent qualities reported for pH, BOD, COD, oil/grease, phenols, and total organic carbon (TOC) are 6–9, <20 mg/L, <200 mg/L, <10 mg/L, <0.25 mg/L, and <75 mg/L, respectively.

**Table 1.** Characteristics of typical petroleum refinery wastewater reported from the literature.

| Parameters | | | | | | | | | | |
|---|---|---|---|---|---|---|---|---|---|---|
| pH | BOD (mg/L) | COD (mg/L) | TSS (mg/L) | TDS (mg/L) | TOC (mg/L) | NH$_3$ (mg/L) | Phenols (mg/L) | Sulfides (mg/L) | Oil and Grease (mg/L) | Reference |
| 7.74 | 155 | 485 | 600 | 800 | - | 13.7 | 3.5 | - | 17.36 | [32] |
| - | 1198 | 2554 | - | - | 610.93 | 81.2 | - | - | - | [33] |
| 6.7 | 174 | 450 | 150 | - | 119 | - | - | - | 870 | [34] |
| 8.3–8.9 | - | 3600–5300 | 30–40 | 3.8–6.2 | - | | 11–14 | - | - | [25] |
| 9.2 | - | 970 | 42.3 | 1220 | - | | | - | - | [35] |
| 7.2 | 107.3 | 232.7 | 86.2 | 276 | - | 0.7 | 0.17 | - | 2.9 | [36] |
| 8.0 | 718 | 1494 | 75 | - | -- | - | 70 | 142 | - | [37] |
| 8.0 | 195 | 480 | 315 | - | - | - | 13.8 | 16.8 | 94 | [38] |
| 8.3–8.7 | - | 3970–4745 | 30–40 | 3800–6200 | | | 8–10 | - | - | [39] |
| 7.82 | - | 310 | - | 1910 | -- | -- | - | - | - | [40] |
| 8.2 | 23 | - | 31 | - | - | 0.81 | 20.7 | - | - | [24] |
| 7.8 | 44,300 | 74,800 | 2010 | 41,600 | 5490 | - | - | - | - | [31] |
| 7.3 | - | 330 | 253.3 | -- | 391 | 9.5 | - | - | - | [41] |
| 7.2 | - | 1179 | - | 74 | -- | - | 257 | 0.18 | 217 | [42] |
| 8.0 | 138 | 350 | 60 | 2100 | - | - | 7.35 | - | 14.75 | [43] |
| 8.0 | 8.6 | 112 | | 930 | - | 0.7 | | - | - | [44] |

**Table 2.** Concentrations of some heavy metal concentrations from typical refinery wastewater.

| | | | | Heavy Metals | | | | | | |
|---|---|---|---|---|---|---|---|---|---|---|
| Cadmium | Chromium | Copper | Lead | Manganese | Iron | Zinc | Arsenic | Mercury | Nickel | Reference |
| <0.005–0.2 | 0.02–1.1 | <0.002–1.5 | <0.004–175 | – | <0.1–100 | 0.01–35 | 0.01–35 | <0.001–0.002 | – | [5] |
| – | <0.01 | <0.01 | 0.04 | 0.58 | 5.14 | 0.75 | <0.4 | <0.15 | 0.02 | [40] |
| 0.045 | 0.022 | – | 0.03 | – | – | – | – | – | 0.176 | [45] |
| ND | – | – | 0.0135 | – | 0.253 | 0.33 | – | – | – | [46] |
| – | 1.225 | 0.005 | 0.47 | – | – | 0.45 | – | – | – | [47] |
| <0.001 | 0.06 | – | – | 0.149 | 2.535 | 1.133 | – | – | – | [48] |
| 0.031 | 2.33 | 0.86 | 2.06 | – | 2.28 | 7.56 | | | 1.03 | [41] |
| 0.054 | 0.025 | | 0.031 | – | 0.775 | 0.75 | – | – | 0.188 | [49] |
| 0.026 | 0.04 | 0.03 | 0.01 | – | 0.88 | 0.03 | | – | – | [50] |
| 5.93 | – | – | – | – | – | – | 2.78 | 1.05264 | – | [51] |

## 3. Treatment of Petroleum Refinery Wastewater

Since petroleum wastewater contains toxic contaminants, which are a major threat to the environmental ecosystem, it is necessary to use the appropriate treatment before disposal and to meet the regulatory requirements. Given this, there are various treatment techniques which have been employed and reported in the literature for the treatment of PRWW [52]. While some already established technologies are efficient in terms of their treatment, cost and energy requirement, others are associated with high energy and maintenance costs and hence are not environmentally friendly. Therefore, efficiency assessment in terms of energy requirements, flexibility to treat various contaminants, and level of waste generation as a by-product at the end of the treatment process is critical to the development and application of any treatment technology [53]. Generally, PRWW treatment has two main stages: the pre-treatment stage, which is used to reduce contaminant loads such as oil, grease, and suspended solids. Secondly, there is degradation of the pollutants to an acceptable discharge limit [3,7,20]. Some reported treatment techniques in the literature include biological processes [25,54–57], coagulation processes [58–60], adsorption processes [25], membrane processes [61–64], chemical oxidation [65], and advanced oxidation processes (AOPS) [66–69]. In most cases, the determination of the treatment efficiency of these techniques focuses on the efficiency of the removal of the BOD, COD, oils and grease, phenols, sulfates, total organic carbon (TOC) as well as the concentration of heavy metals. Based on this, advanced oxidation processes such as Fenton oxidation and photocatalysis are receiving more attention nowadays due to their high capability to delete recalcitrant petroleum contaminants [7]. Many advances in treatment technologies have been achieved in recent years due to advancements in material science caused by a dynamic approach to the treatment of emerging contaminants [70]. In this review article, a review of the works previously reported using conventional as well as advanced and integrated treatment techniques for the treatment of PRWW will be discussed (Figure 5).

### 3.1. Conventional Treatment Techniques

PRWW effluent can be treated using either conventional, advanced, or integrated treatment processes (Figure 6). Conventional techniques have been widely used since the beginning of the 20th century and mainly consist of a combination of physical, chemical, and biological processes. According to Yu et al. [4], conventional techniques for the treatment of PRWW include flotation, coagulation, biological treatment, and membrane separation technology. However, these techniques are usually associated with various limitations including low efficiency, high capital operating cost as well as low sensitivity to emerging complex organic contaminants [71]. Toxic recalcitrant pollutants from hydrocarbon sources such as naphthenic acids (NAs) usually remain a considerable challenge in the treatment of PRWW using biological processes. Furthermore, due to their low efficiency and operational limitations, it makes it necessary to adopt more robust advanced treatment systems to achieve MCL. It usually includes a sequence of mechanical and physicochemical processes, followed by biological treatment of usually activated sludge treatment units. Some of the

notable biological treatment systems in this regard, as indicated in Figure 6, include up-flow anaerobic sludge blankets (UASB), sequence batch reactors (SBR), membrane bioreactors (MBR), up-flow anaerobic fixed beds (UAFB), granular sludge beds (EGSB) and anaerobic baffled reactors (ABR).

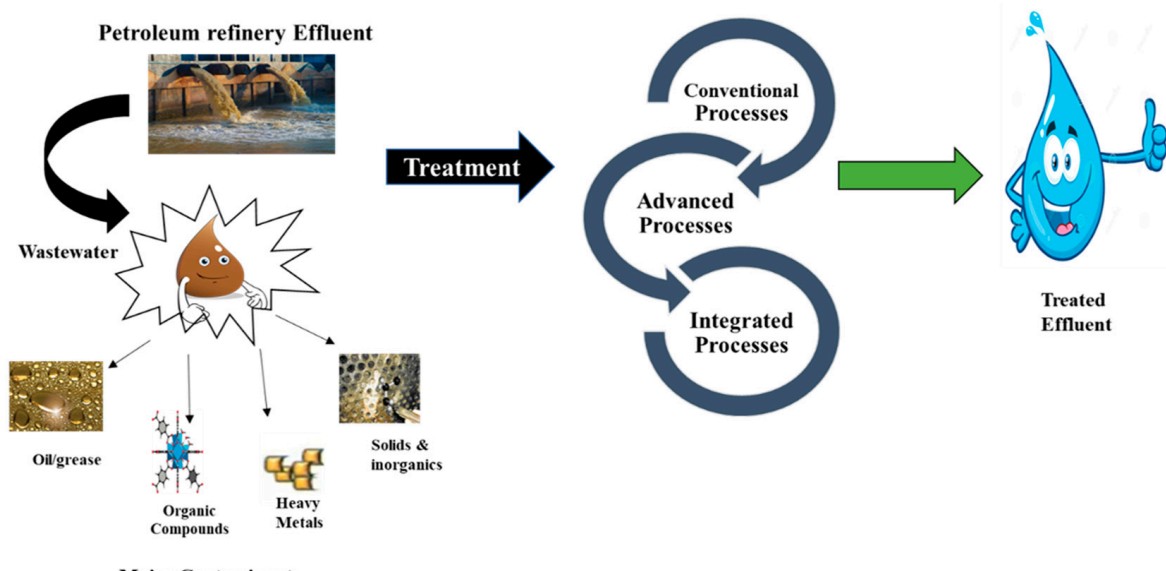

**Figure 5.** Composition of petroleum refinery wastewater and major classifications of treatment techniques.

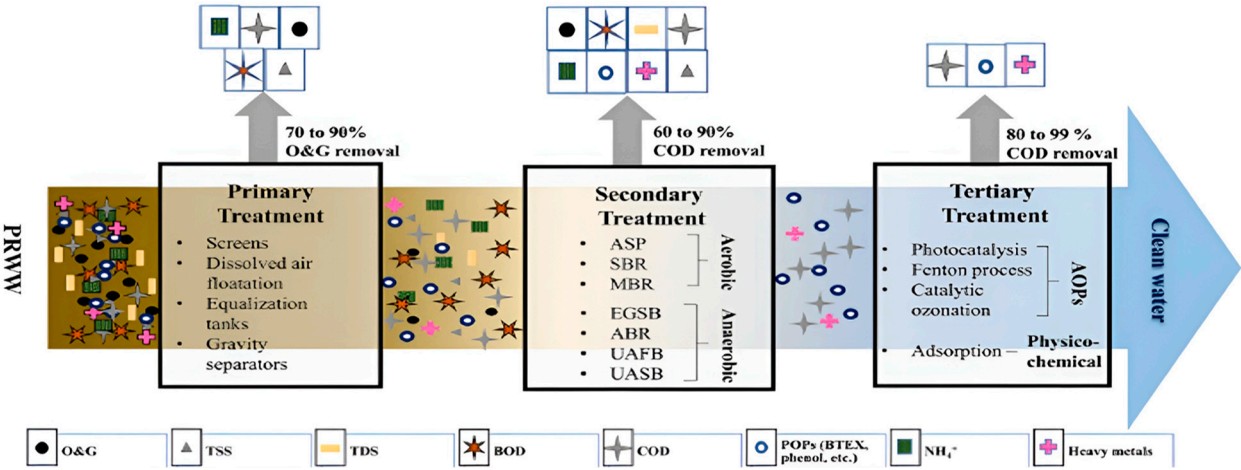

**Figure 6.** General overview of the PRWW treatment techniques [8].

The pre-treatment stage is sometimes regarded as part of the primary treatment, where the majority of the suspended solids are separated and removed with the help of gravity, sedimentation, and filtration processes [29]. The advanced treatment systems include advanced oxidation processes, (such as photocatalysis, Fenton-oxidation, and electrochemical processes) and have been reported to provide more efficient treatment and less production of by-products that may also require further treatment. Alternatively, the use of an integrated or hybrid system which combines two or more advanced processes is also, nowadays, receiving more attention to provide the most effective treatment for the removal of oil and other hazardous pollutants from petroleum wastewater [9].

### 3.1.1. Physicochemical Processes

The physicochemical processes are a set of techniques that employ the use of the physical and chemical properties of the PRWW in the removal of contaminants. The conventional physical techniques usually include the use of filtration, floatation, adsorption, and sedimentation, while the chemical techniques include precipitation and coagulation processes [7]. The physical PRWW treatment process techniques such as screening, floatation, sedimentation, and gravity separation do not require the application of biological or chemical agents during the treatment process. They usually constitute the primary treatment process to reduce the waste load before proceeding to the secondary treatment units [7].

#### Flotation and Sedimentation Processes

The sedimentation process is used for the separation of water and oil due to the density difference. Hence, a significant density difference is required to provide an optimum separation. Oil and water sedimentation can be mechanically achieved using separators such as the API separator which operates on the principle of specific gravity differences to allow the settlement of heavy oil and pollutants [71]. Diffused air flotation (DAF) is achieved by introducing fine air bubbles to enhance the formation of a scum layer between the oil and the water for easy separation. The technique is achieved by introducing air under pressure which results in the pollutants rising to the top surface (Figure 7). High levels of total suspended solids, colloids, as well as some immiscible liquids are significantly reduced during this stage [72]. Abuhasel et al. [69] reported that DAF techniques enhanced by nanobubble systems were applied along with surfactants to reduce the surface tension of the oil concentration. About 90% oil separation efficiency was reported using this system. The technique has shown better efficiency than the traditional DAF system. Floatation and gravity separation were usually used as the first stage separation process to remove floating and dispersed oil efficiently. However, they are not efficient in terms of the separation of emulsified oil [70,73]. Li et al. [74] used the application of a diffused floatation process to a sedimentation tank of PRWW with an effluent oil concentration of 3000–14,000 mg/L. An average effluent oil concentration of 300 mg/L with a minimum value of 97 mg/L was achieved using this process. However, according to Wang et al. [75] most conventional physicochemical techniques, especially floatation and sedimentation processes, do not yield more than 16% to 24% efficiency in the removal of organic aromatic pollutants. Furthermore, high maintenance costs and increased energy consumption were also among the major disadvantages of the DAF system.

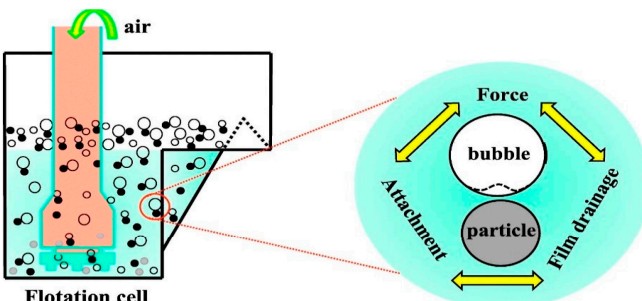

**Figure 7.** The mechanism of dissolved air flotation [70].

#### Coagulation/Flocculation

The use of coagulants to form flocs is also another physicochemical treatment process for the removal of pollutants from PRWW. Iwuozor [76] reported that coagulants are generally polyelectrolytes or synthetic organic polymers with high molecular weight which form multi-charged poly nuclear complexes in solution that makes flocs settle easily (Figure 8). The function of the coagulant is to promote the agglomeration or accumulation of wastewater particles by reducing the surface charges of the electrostatic particles. Different investigations have indicated the ability of the coagulation process for the treatment of

PRWW. Coagulation processes were reported to be effective in treating heavy metals and high-level concentrations of organic pollutants (Table 3). Hassan et al. [77] reported that coagulation/flocculation are the most popular techniques for the removal of pollutants related to turbidity, colour, and total suspended solids (TSS). However, it is an inappropriate technology for the complete removal of organic pollutants but can be efficiently applied before membrane and biological processes to reduce the level of non-biodegradable pollutants [78]. Although it is greatly influenced by pH, coagulant dose and settling time, the coagulation process is also reported to remove dissolved and emulsified oils [79]. Aluminium and iron salt coagulants such as aluminium sulfate (alum), ferrous sulfate and ferric chloride were among the most widely used coagulants [80,81]. Zueva et al. [60] reported research conducted to prove the efficiencies of $Ca(OH)_2$ and $Al_2(SO_4)_3$. Under optimum conditions, the removal efficiencies of turbidity, total hydrocarbons and COD were 100%, 90% and 70%, respectively. Similarly, Ayhan et al. [80] also reported the application of $Al_2(SO_4)_3 \cdot 18H_2O$ and obtained efficiencies of 78.75 and 98.10% for COD and turbidity removal at a pH of 9.43. On the other hand, the treatment capacity of potassium ferrate ($K_2FeO_4$) in very oily wastewater was tested by Kareem and Tameemi [81] and reported to have about 93.50% COD removal efficiency. From these studies, it can be concluded that these metal salt coagulants are good for the treatment of PRWW with organic compounds. Moringa oleifera and alum combination as coagulants have also been reported to reduce turbidity and TSS by 62.16 and 61.05%, respectively. The efficiency of coagulants can be modified with the use of additives which can enhance their ability during treatment processes. Hassan et al. [82] have reported the use of ferric sulfate coagulant and polyelectrolyte (polyacrylamide) as an additive. At an optimum pH of 6.86, about 86.67% oil removal efficiency was achieved. Hence, this study proved that the efficiency of ferric sulfate coagulant can be improved with organic compound additives.

**Table 3.** Petroleum refinery wastewater treatment by coagulation.

| S/No. | Adsorbent | Experimental Conditions | | | | Pollutants Removed | Removal Efficiency (%) | Reference |
|---|---|---|---|---|---|---|---|---|
| | | pH | Dosage | Temp. (°C) | Time (Min) | | | |
| 1 | Activated carbon (AC), natural clay (NC) and sawdust (SD) | 7 | NC 18.96 mg/g, AC 16.25 mg/g & SD 14.11 mg/g. | NR | 100 | Colour | 83.1 | [83] |
| | | | | | | COD | 67.2 | |
| 2 | Activated carbon fixed-bed column | 5.7 | 80% Parking | 25 ± 2 | 73 | COD | 96.7 | [84] |
| 3 | Synthesized nanorods ZnO/SiO₂ via the sol-gel | | | | | $Pb^{2+}$ | 85.06 | [85] |
| | | | | | | $Cd^{2+}$ | 84.12 | |
| 4 | Functionalized mesoporous material with amine groups (NH₂-MCM-41) | 7 | 0.4 g/L | | 50 | PAHs | 85.7 | [86] |
| 5 | ZnO/Fe₃O₄ nanocomposite | NR | 0.08 g | 30 | 900 | $Cu^{2+}$ | 92.99 | [87] |
| | | | | | | $Cr^{6+}$ | 77.60 | |
| 6 | Date pit-activated carbon (DPAC) | | | | | COD | 95.0 | [87] |
| | Wooden activated carbon | 8–9 | 2 g/L | | | BOD | 95.00 | [88] |
| | | | | | | TOC | 88.00 | |
| | | | | | | COD | 68.67 | |
| 7 | Graphene oxide nanocomposites with Cadmium oxide (CdO) | 5 | 1.3 g/L | | | $Cr^{6+}$ | 98 | [89] |

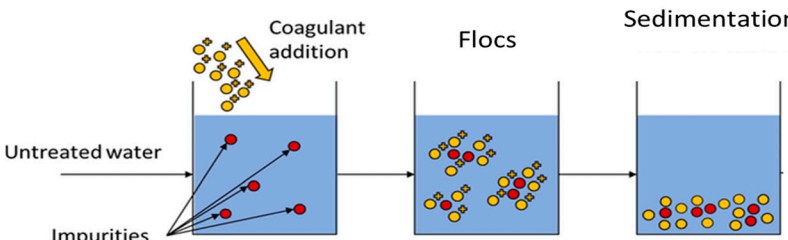

**Figure 8.** Illustration of the mechanisms of coagulation, flocculation and sedimentation [90].

Adsorption Using Conventional Adsorbents

The adsorption process can be both a conventional and advanced treatment technique depending upon the adsorbent material as indicated in Figure 9. Conventional adsorbent materials such as activated carbon, zeolites and silica have been used for a long time in the treatment of PRWW. Nowadays, adsorption techniques (using both conventional and non-conventional adsorbents) are one of the commonly studied techniques for industrial wastewater treatment due to their simplicity and lower treatment cost [91]. Additionally, besides its effectiveness and economic advantage, the adsorption technique is sometimes a reversible process where adsorbents can be regenerated simply through an appropriate desorption process [92]. Parameters affecting adsorption process efficiency include pH, temperature, contact time and adsorbent porosity and dose. The pH factor affects the ability of the hydrogen ions in the solution, their interaction with the functional groups and the metal ions in the case of heavy metals [93].

Adsorbents with high porosity tend to have a high surface area as well as high adsorption capacity [94]. Activated carbons, polymeric organic resins (such as ion exchange resins) and inorganic adsorption materials such as zeolites, silica gel, as well as activated alumina, were all classified as conventional adsorbents. Meanwhile, industrial, or agricultural by-products such as rice husk and sawdust were categorized as non-conventional adsorbents [95]. Various adsorbent materials derived from agricultural or industrial by-products originating from natural materials or modified biopolymers were reported to be used for the removal of heavy metals [96]. Fadali et al. [83] have reported the application of activated carbon, natural clay, and sawdust for the treatment of petroleum wastewater from the Kuwait Gulf Oil Company (KGOC) in Al Ahmadi, Kuwait. The sorption capacities reported were 15.52 mg/g, 16.23 mg/g and 12.91 mg/g for the activated carbon, natural clay, and sawdust, respectively, at 100 min of equilibrium time. The sorption capacity of an adsorbent is always related to the material's surface chemistry which describes its available pore size that would accommodate contaminants during the adsorption process [97]. Wang et al. [88] investigated the adsorption potential and efficiency of palm kernel shells (PKS) as biomass integrated with iron oxide and zeolite. The analysis revealed that the optimized PKS can remove colour (83.1%) and COD (67.2%) within a contact time of 30 min. Similarly, Kassob and Abbar [84] also investigated the COD removal efficiency of an activated carbon fixed-bed column operated in a batch recirculation mode using petroleum wastewater from Iraq's Al-Diwaniyah petroleum refinery plant. At an optimal pH of 5.7 with about 80% activated carbon column packing, a 96.70% COD removal efficiency was recorded after 76 min. Bukhari et al. [89] have reported the composite adsorption ability of graphene oxide nanocomposites with cadmium oxide (CdO). At an adsorbent dosage of 1.3 g/L and pH 5, a $Cr^{+6}$ efficiency of 98% was obtained. As a nanomaterial, the adsorbent showed effective adsorption up to 85% even after five adsorption cycles. On the other hand, Wang et al. [88] also reported the application of wooden activated carbon for the separate and combined applications of adsorption and coagulation. From their research, the reported BOD and TOC removal efficiencies were 95 and 88%, respectively. Based on their comparison, their adsorbent has shown greater adsorption capacity than the natural zeolite adsorbent.

Various adsorbent materials employed for the treatment of PRWW have been summarized in Table 4.

**Table 4.** Petroleum refinery wastewater treatment by adsorption.

| S/No. | Adsorbent | pH | Dosage | Temp. (°C) | Time (Min) | Pollutants Removed | Removal Efficiency (%) | Reference |
|---|---|---|---|---|---|---|---|---|
| | | | | **Experimental Conditions** | | | | |
| 1 | Activated carbon (AC), natural clay (NC) and sawdust (SD) | 7 | NC 18.96 mg/g, AC 16.25 mg/g and SD 14.11 mg/g | NR | 100 | Colour | 83.1 | [83] |
| | | | | - | - | COD | 67.2 | |
| 2 | Activated carbon fixed-bed column | 5.7 | 80% Packing | 25 $\pm$ 2 | 73 | COD | 96.7 | [84] |
| 3 | Synthesized nanorods ZnO/SiO$_2$ via sol-gel | | | - | - | Pb$^{2+}$ | 85.06 | [85] |
| | | | | - | - | Cd$^{2+}$ | 84.12 | |
| 4 | Functionalized mesoporous material with amine groups (NH$_2$-MCM-41) | 7 | 0.4 g/L | - | 50 | PAHs | 85.7 | [86] |
| 5 | ZnO/Fe$_3$O$_4$ nanocomposite | NR | 0.08 g | 30 | 900 | Cu$^{2+}$ | 92.99 | [87] |
| | | | | - | - | Cr$^{6+}$ | 77.60 | |
| 6 | Date pit-activated carbon (DPAC) | | | - | - | COD | 95.0 | [89] |
| 8 | Wooden activated carbon | 8–9 | 2 g/L | - | - | BOD | 95.00 | [88] |
| | | | | - | - | TOC | 88.00 | |
| | | | | - | - | COD | 68.67 | |
| 9 | Graphene oxide nanocomposites with Cadmium oxide (CdO) | 5 | 1.3 g/L | - | - | Cr$^{6+}$ | 98 | [89] |

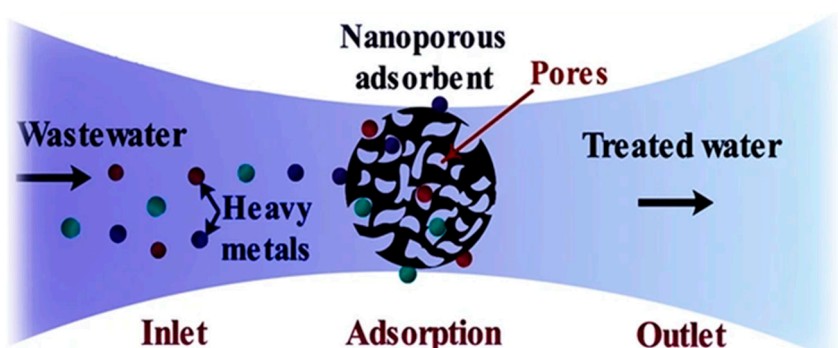

**Figure 9.** The process of adsorption using a nano-porous adsorbent for heavy metals [98].

Membrane Processes

The application of membrane technology for the treatment of wastewater has been in existence since the 18th century [99]. It is a physicochemical treatment technique that is gaining more acceptance nowadays and is also efficient in the treatment of organic matter. Membranes are used as a selective barrier (Figure 10) for the separation of two phases through a semi-permeable pore space by the restriction of movement between components [14]. The membrane mechanism in Figure 10 has shown the successful separation of the coloured contaminations represented in brown over the membrane surface. According to Ezugbe [14] and Aljuboury et al. [7], membranes can be generally classified into two main types; organic membranes (usually made from organic polymers) and inorganic membranes (made from silica, metals, zeolites, or ceramics). Depending on their pore sizes, membranes can be used for microfiltration ultrafiltration or nanofiltration [100].

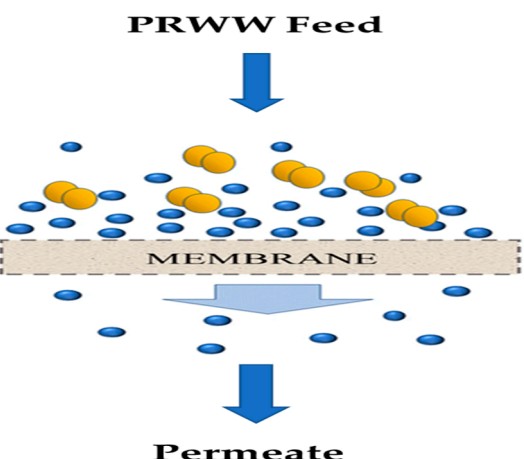

**Figure 10.** Illustration of the mechanism of a membrane process [101].

A high level of concentrate generation which subsequently leads to membrane fouling is the major drawback in the membrane treatment systems [102]. A summary of the reported literature using the membrane process for PRWW is presented in Table 5. Ratman et al. [61] reported the application of a polyether sulfone (PES) membrane consisting of zinc oxide (ZnO) nanoparticles followed by UV irradiation for the pre-treatment of PRWW. The result showed pre-treatment enhanced rejection up to 18.6%, 16.7%, and 87.1%, for total dissolved solids (TDS), COD, and ammonia, respectively. Similarly, Hashemi et al. [45] used micellar-enhanced ultrafiltration (MEUF) for the treatment of heavy metals at Kermanshah Oil Refinery and the results showed 96%, 95%, 92% and 86% removal efficiency for nickel, lead, cadmium, and chromium respectively. Sodium dodecyl sulfate (SDS) was used as a surfactant and added to the effluent to enhance complex formation that can trap heavy metals. Similarly, Madaeni et al. [103] have reported a study using a Polysulfone-Nano $TiO_2$ Hybrid membrane coupled with an ozonation process as a pre-treatment for removing TDS, COD, and phenols. The ozonation process enhanced the membrane permeate capacity by up to 96% and improved the pollutant removal efficiency by up to 77%. They further reported that the ozonation process also reduces the fouling of the membrane and increases surface resistance by up to 21%. However, ozonation is costly, utilizing a hybrid process that can reduce the concentration of pollutants before membrane filtration could enhance membrane efficiency while reducing fouling. Similarly, some membrane technologies are configured along with bioreactors in treating PRWW. For example, a hollow-fibre membrane bioreactor (HF-MBR) was tested for treating a real PRWW sample by Razavi and Miri [63]. The ultra-filtration unit of the membrane yielded 82, 89, and 99% removal efficiencies for COD, BOD and TSS, respectively. This removal efficiency was obtained after running the membrane for 160 days. Similarly, Lebron et al. [104] have reported the performance of a hybrid ultrafiltration-osmotic membrane bioreactor (UF-OMBR) to treat oily refinery wastewater. The dissolved organic carbon (DOC) removal efficiency is about 99% compared to 66% for the ordinary conventional MBR systems. A laboratory scale testing for the comparison of hydrophilic microfiltration (MF GRM) and ultrafiltration (UF GRM) polymeric membranes was conducted by Asatekin and Mayes [105] for the treatment of oily wastewater effluent. A similar removal capacity of about 99% was observed for both membranes. Although high removal efficiency is achieved with this system, the problem of increasing salinity in the MBR unit is usually the main challenge associated with osmotic membranes. Generally, the major limitation connected with bioreactors is their long contact time compared to organic/inorganic membranes. To avert the problem associated with salinity build-up, Lebron et al. [104] have used magnesium chloride ($MgCl_2$) as a draw solution within the mixed liquor. This helps to enhance nitrification and denitrification processes thereby increasing the permeate efficiency. It can be noted that various membranes have been reported for the treatment of PRWW. Kusworo et al. [106] utilized a Polysulfone (PSf) membrane with improved efficiency using zinc oxide (ZnO) nanoparticles and reported rejection

values of 70.21% and 74.68% for TDS and COD, respectively. Most of these membranes were produced from an organic polymeric substance such as polysulfone or cellulose. Occasionally, membranes are also produced from an inorganic material such as alumina and glass with a high molecular weight [107].

**Table 5.** Petroleum refinery wastewater treatment by membrane processes.

| S/No. | Membrane | Pollutants Removed | Removal Efficiency (%) | Reference |
|-------|----------|--------------------|------------------------|-----------|
| 1 | Polyether sulfone (PES) membrane consisting of zinc oxide (ZnO) nanoparticles | TDS | 18.6 | [61] |
| | | COD | 16.7 | |
| 2 | Micellar-enhanced ultrafiltration (MEUF) | Ammonia | 87.1 | [45] |
| | | Nickel | 96 | |
| | | Lead | 95 | |
| | | Cadmium | 92 | |
| | | Chromium | 86 | |
| 3 | Polysulfone zinc oxide (ZnO) nanoparticles to PSf membrane | TDS | 70.21 | [64] |
| | | COD | 74.68 | |
| 4 | Polysulfone-nano $TiO_2$ hybrid membrane | TDS | 77% | [108] |
| | | COD | 77.2 | |
| | | Phenols | 78.5 | |
| 5 | HF-MBR | COD | 82 | [63] |
| | | BOD | 89 | |
| | | TSS | 99 | |
| 6 | MF GRM and UF GRM polymeric membranes | DOC | 99 | [104] |
| 7 | Polyvinyl chloride–titanium oxide (PVC–$TiO_2$-NPs) membranes | COD | 79.6 | |
| 8 | Polyacrylonitrile-graft-poly (ethylene oxide) UF membranes | COD | 96% | [107] |
| 9 | A sheet nano-porous membrane (PAN) | TSS | 100 | [106] |
| | | TDS | 44.4 | |
| | | Oil/grease | 99.9 | |
| | | COD | 80.3 | |
| | | BOD | 76.9 | |

*3.2. Chemical Processes*

Chemical Precipitation and Ion Exchange

Chemical processes utilize the application of chemical reactions in the removal of contaminants from wastewater. Neutralization, ozonation, ion exchange and oxidation processes are among the most widely used chemical processes in the treatment of PRWW [32]. Neutralization consists of the use of an acid or base such as lime to adjust the pH level [7]. Generally, chemical precipitation is one of the most widely used conventional chemical processes for the removal of heavy metal concentrations from inorganic effluents [45]. In the precipitation process (Figure 11), heavy metal ions react with suitable chemicals called precipitants to form insoluble precipitates. which can be further separated by a sedimentation or filtration process [104]. It is a relatively simple and less costly technique which can be used for the removal of metals and sulfides. The coagulation precipitation method is broadly used with the help of chemical precipitants such as $Ca(OH)_2$ and NaOH as indicated in equation 1 [98]. Alnakeeb and Rasheed [109] have reported the application of $BaCl_2$ and Al $(OH)_3$ in the treatment of PRWW from Al-Doura Refinery in Iraq. High sulfate removal efficiency was obtained with $BaCl_2$ over Al $(OH)_3$ and it was concluded that aluminium hydroxide is unsuitable for PRWW with neutral pH and low sulfate concentrations. Barium salts are highly insoluble making them an excellent precipitant for sulfate ions. Altaş and Büyükgüngör [110], also reported the use of $Ca(OH)_2$ as a precipitant modified with $Fe^{2+}$ ions and obtained 96–99% and 50–80% removal efficiencies for sulfide and COD, respectively. Alternatively, precipitation can also be achieved using sodium or

calcium carbonates in which classical carbonates are formed. Habte et al. [111] investigated the removal efficiency of cadmium and lead via carbonation of aqueous $Ca(OH)_2$ derived from eggshell and found the results to be efficient for obtaining very low concentrations of heavy metals. About 99.99% and 99.63% treatment efficiency for $Cd^{2+}$ and $Pb^{2+}$ were achieved at an optimum dosage of 3 g/L of $Ca(OH)_2$, with an initial metal concentration of 100 mg/L and a $CO_2$ flow rate of 1 L/min. The study provided evidence for the application of $Ca(OH)_2$ derived from eggshells for the treatment of heavy metals. On the other hand, the use of carbonation to enhance $Ca(OH)_2$ based precipitation can be an attractive method to enhance the capture and utilization of $CO_2$ as a greenhouse gas. However, the formation of large amounts of sludge and the effect of pH is the main disadvantage of the precipitation process [112]. Furthermore, high concentrations of chlorides also affect the performance of the precipitation process. This is because at higher chloride concentrations the formation of hydroxy salt precipitates is favoured instead of the typical heavy metal hydroxides [113]. From their review of about 185 articles from 1988–2010, Fu and Wang [114] stated that ion exchange, adsorption and membrane filtration were the most widely studied methods for the treatment of heavy metals. A summary of some precipitation techniques employed is given in Table 6. The potential recovery of the metal, higher selectivity, and lower sludge production are among the main advantages of the ion exchange technique. The main principle of the technique is the exchange of ions in a chemically equivalent amount between a resin (usually a solid) and an electrolytic solution [97]. Generally, ion-exchange resins are applied for the isolation of rare metals for the regeneration of metal wastes as well as a softening process [115]. The resin materials can be natural such as inorganic zeolites or synthetically produced organic resins [98].

$$Pb\,(NO_3)_2 + 2NaOH \rightarrow 2NaNO_3 + Pb\,(OH)_2\downarrow \tag{1}$$

Equation of the reaction for the chemical precipitation of lead metal ions using sodium hydroxide.

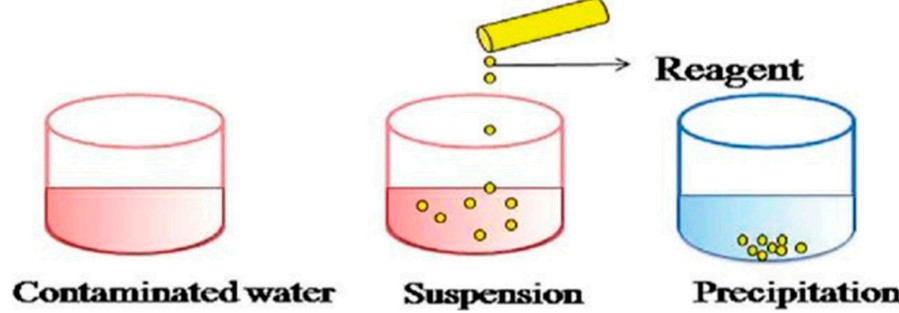

**Figure 11.** Illustration of a chemical precipitation process of lead metal ions [116].

**Table 6.** Treatment of petroleum refinery wastewater by precipitation.

| S/No. | Precipitant | pH | Dosage | Experimental Conditions Temp. (°C) | Time (Min) | Pollutants Removed | Removal Efficiency (%) | Reference |
|---|---|---|---|---|---|---|---|---|
| 1 | $BaCl_2$ and $Al\,(OH)_3$ | 7 | 0.36 g/L | NR | 15 | Sulfate ion | | [109] |
| 2 | $Ca(OH)_2$ and $Fe^{2+}$ ions | 5 | 40 mg/L | NR | NR | Sulfide | 97.5 | [110] |
| | | | | | | COD | 65 | |
| 3 | $Ca(OH)_2$ derived from eggshell | NR | 3 g/L | NR | NR | $Cd^{2+}$ | 99.99 | [111] |
| | | | | | | $Pb^{2+}$ | 99.63 | |

### 3.3. Biological Processes

Biological processes utilize the use of microbial activity from living organisms such as bacteria to decompose or degrade organic contaminants. The four major groups of

biological treatment processes are aerobic, anaerobic, anoxic (the process by which nitrate is biologically converted into nitrogen gas in the absence of oxygen), or a combination of the three. The principal applications for these processes are removing carbonaceous organic matter (measured in BOD, COD, or TOC), nitrification, denitrification, or stabilization [114]. There are various biological processes which have been reported to be effective in treating PRWW, among which the activated sludge process is the most widely used [5]. However, there is little removal efficiency of petroleum hydrocarbons and a large amount of sludge production, but at least up to 60–90% COD removal efficiency was observed in many biological treatments of PRWW [8].

Furthermore, anaerobic digestion produces methane gas as a renewable energy and requires less space and sludge than the aerobic process. Before the application of the biological treatment techniques, pretreatment processes such as flotation, flocculation, sedimentation, and filtration are usually employed to eliminate the free oil and gross solids as well as increase biodegradability [117]. Different reactor systems for the aerobic process have been designed to treat PRWW including the traditional activated sludge system, contact stabilization active sludge, membrane bioreactor (MB), biological aerated filter (BAF), moving bed biofilm reactor (MBBR), sequence batch reactor (SBR) etc. Bioreactors adopted for PRWW are generally categorized as suspended growth, attached growth or hybrid process treatments [118]. Shuokr and Sazan [71] have reported that a high COD removal efficiency of up to 78% and 94% for TOC and oil degradation was achieved using an aerobic system. Rasheed and Muthukumar [119] also reported the treatment of a real PRWW sample with an initial COD of 40,000 mg/L and a pH of 5.4 using a sequencing batch bioreactor (SBR) with sonication for 30 min as pre-treatment. Their investigation revealed a significant decrease in the COD with an increase in time. An industrial-scale granular sludge bed bioreactor and aerobic-activated sludge treatment (EGSB-BR) were developed by Liang et al. [120] to treat PRWW. The total COD and petrochemical removal efficiencies of the plant were 85.6 % and 81.5 %, respectively. El-Naas et al. [39] investigated a three-step pilot plant process consisting of biological treatment in a spouted bed bioreactor (SBBR) unit for the treatment of highly contaminated PRWW in which they achieved 96% COD removal and nearly 100% degradation of phenols. Furthermore, Vendramel et al. [56] utilized the capability of an aerobic submerged fixed-bed reactor (ASFBR) to treat a high organic strength PRWW and found COD, dissolved organic carbon and TSS removal efficiencies of 91%, 90% and 92%, respectively. About 90% reduction in the ammonium level was also obtained. However, almost all biological treatment techniques have been associated with the common major limitation for large sludge generation and the inactivity of microbial organisms to toxic recalcitrant. The investigations presented in Table 7 have indicated their efficiency in the reduction of organic contaminants from PRWW. It can be observed that most studies do not provide phenols and TOC removal efficiencies.

**Table 7.** Treatment of petroleum refinery wastewater by biological processes.

| S/No. | Biological Process/Reactor | COD (%) | TOC (%) | Phenols (%) | TSS (%) | Reference |
|---|---|---|---|---|---|---|
| 1 | Aerobic biological treatment | 78 | 94 | | | [72] |
| 3 | Granular sludge bed bioreactor and aerobic-activated sludge treatment (GSB-BR) | 85.6 | NR | NR | NR | [120] |
| 4 | Spouted bed bioreactor (SBBR) | 96 | NR | 100 | NR | [39] |
| 5 | Aerobic submerged fixed-bed reactor (ASFBR) | 91 | | NR | 92 | [56] |
| 6 | Membrane bioreactor (MBBR) | 80 | NR | NR | NR | [121] |
| 7 | Up-flow anaerobic sludge blanket (UASB) reactor biological aerated filter (BAF) | 90.2 | NR | NR | NR | [122] |
| 8 | Microbial fuel cells (MFCs) | 63.1 | NR | NR | NR | [123] |
| 9 | Multi-stage biological reactors (MSBR) | 98% | NR | NR | NR | [124] |
| 10 | Rotating Biological Contactor (RBC) | 87 | 55 | 99 | 85 | [125] |

### 3.3.1. Bioremediation Using Constructed Wetlands

The PRWW pollutants can also be combated using plant accumulation capabilities in the form of constructed wetlands. Phytoremediation is where plants alone and their associated microorganisms are used to degrade pollutants from contaminated systems. Based on this, constructed wetlands are widely used as a major technology in the restoration of oil-polluted environments to restore natural habitats [118]. Unlike physical and chemical processes, bioremediation is seen as a more environmentally friendly system due to its generation of less hazardous reaction products. Jain et al. [8] reported the performance of horizontal subsurface flow-constructed wetlands for the treatment of PRWW and petrochemical plant wastewater. They revealed that horizontal subsurface flow has a better performance, of about 80% and 90% efficiency, to remove heavy oil and recalcitrant organic compounds. As an alternative technology, different phenolic compounds, even at high concentrations, can be effectively removed using constructed wetlands [126]. Although it seems an environmentally wise technique, but it requires a very large space to construct a successful wetland to treat industrial effluent of a petroleum refinery. Hence, this indicates a major drawback to the successful implementation of wetlands.

### 3.4. Advanced Treatment Processes

The problem of low treatment efficiency and high operational costs among others in most conventional treatment processes have led to the need to adopt more advanced treatment technologies. Alternatively, the application of hybrid systems which combine the use of two or more techniques is many times more effective for the removal of oil and other hazardous contaminants from PRWW [9].

### 3.4.1. Adsorption Using Modified Adsorbents

With the advancements in the field of material science and the need for an effective and low-cost adsorbent, different natural and synthetic materials have been tested for the adsorption of contaminants from wastewater of different industrial effluents. Although the selection of an appropriate adsorbent material with suitable properties is indispensable in obtaining maximum adsorption capacity, the adsorption technique is often seen as the best choice in the treatment of different types of wastewater. Vikrant and Kim [127] maintained that this is because the adsorption technique is regarded as the simplest and most fitting treatment technique for almost all types of wastewater [128]. Additionally, the adsorption technique is sometimes also believed to be the optimal method even for crude oil spill clean-up because of its relatively low cost and high efficiency. Various oil hydrophobic adsorbents exist nowadays, such as natural sorbents, synthetic organic polymers, and inorganic mineral materials generated from a variety of sources that can be used to treat oily PRWW [128]. For example, Abdeen and Moustafa [129] have reported their study for the adsorption of crude oil from PRWW on a crosslinked polyvinyl alcohol (PVAH) and its foam structure (PVAF). The macro-porous adsorbent of PVAF was prepared by adding $CaCO_3$ and epichlorohydrin, which act as the pore-forming agent and crosslinker, respectively. The adsorption ability of the two materials was assessed using the gravimetric method where the PVA-F carrier demonstrated an improvement in hydrocarbon trapping over the ordinary PVA. The crude oil removal ability of the PVA-F was approximately 82% at a pH of 3. Meanwhile, the removal percentage is higher at pH 3 and 9 compared with pH 7. This study confirms the potential ability of using PVA hydrogels in the form of PVA-F for oily PRWW treatment, especially in an open marine environment. Furthermore, it also proved the good ability of calcium carbonate as a pore-forming agent in the preparation of hydrogel adsorbents. However, there is a need for an optimum pH determination for the effective use of the hydrogels for the treatment of PRWW. Similarly, Li et al. [130] also reported the use of a hydrogel composite produced by a freeze–thaw process using chitosan, polyvinyl alcohol, and carbon black as raw materials and applied to oil/water separation. The prepared hydrogel displayed an efficient oil repellence and water affinity properties in the separation of oil/water mixtures. After 25 oil–water separation cycles,

the hydrogel-coated filter still had a separation efficiency of over 98%. Furthermore, they also reported that due to its super hydrophilicity and active functional groups, it was able to effectively absorb dye molecules dissolved in water. Hydrogels are sometimes prepared in the form of films or sponges. Li et al. [130] similarly reported the synthesis of a highly hydrophobic and self-recoverable hydrogel sponge prepared from cellulose nanofibrils (CNFs), N-alkylated chitosan (NCS), and poly (vinyl alcohol) (PVA) for oil/water separation. The interconnected microstructure CNF/NCS/PVA hydrogel was found to have 96% porosity. The hydrogel sponge effectively separates oil/water mixtures and water-in-oil emulsions with high separation efficiency and good stability in various acidic, saline, and mechanical conditions. They further maintained that it could absorb various organic solvents with an absorption capacity of about 19.05–51.08 times its original weight. Similarly, Xue et al. [131] have conducted a study to separate an oil/water mixture in highly acidic, alkaline, and salty conditions using a porous calcium alginate/silver nanoparticle (Ca-ALG/Ag) hydrogel film with super hydrophilic and underwater superoleo phobitic properties which are fabricated through an eco-friendly process. The synthesis of the Ca-ALG hydrogel film was conducted by combining ionic cross-linking of $Ca^+$ ions and a soluble NaCl salt-template method and incorporating the Ag nanoparticles into the alginate matrix by a simple reduction process. NaCl crystals were used as templates and sifted on the ALG solution films which can be easily removed by water. The formation of the film was achieved by quickly immersing the ALG/NaCl composites into a solution of $CaCl_2$ solution and sonicating at the same time. The NaCl crystals pierced through the ALG film and dissolved in water gradually generating a macro-pore structure. They finally reported that the oil/water separation efficiency of the Ca-ALG/Ag hydrogel film was above 98%. Polyvinyl alcohol and formaldehyde hydrogel composite sponges (PVF/PVF) were also synthesized from a study conducted by Zheng et al. [132] for the treatment of oily wastewater. Although the prepared hydrogel sponge showed almost 100% oil removal efficiency, it could effectively only remove oil emulsions under a gravity effect with a maximum flux of $2.9 \times 10^5$ L m$^{-2}$ h$^{-1}$ bar$^{-1}$. The hydrogel has displayed excellent reusability and is recovered simply by washing. Tai et al. [133] reported the development of a superhydrophobic composite aerogel-prepared leached carbon black waste (LCBW) obtained from industrial waste and polyvinyl alcohol (PVA) via conventional freeze-casting followed by a surface coating. The synthesized PVA/LCBW aerogel was used as a selective adsorbent for different oils and organic solvents and showed an adsorption capacity of about 35 times its original weight. It can also be reused repeatedly and recovered easily through a simple washing and drying process. The maximum removal efficiency was obtained from a combination ratio of 1:0.5 wt % PVA/LCBW. This corresponds to the highest water contact angle of $156.7 \pm 2.9°$. Meanwhile, for a successful oil/water emulsion separation using porous materials such as hydrogels, the two most important key points for consideration are (1) proper average pore size and (2) the wettability of the adsorbent. This important property describes the level of hydrophilicity and hydrophobicity of the adsorbent material. The superoleophobicity and wettability of the hydrogel adsorbents protect them from fouling by oils, thus making them perform better in removing oil concentrations and reusing [134]. Sha et al. [135] developed polyvinyl alcohol–formaldehyde (PVA–PVF) sponges with harmonious pore size through a crosslinking reaction of polyvinyl alcohol (PVA) in the polyvinyl alcohol–formaldehyde (PVF) and under acidic conditions. They further stated that the use of PVA containing chitosan, diatomite, and sodium alginate (SA) can effectively decrease the average pore size of PVF from approximately 75 μm to 23 μm along with a few hundred-nanometer pore channels while maintaining porosity above 73.4%. The oil/water emulsion separation efficiency can reach up to 97.40% with a high-water flux of $2.40 \times 10^4$ L m$^{-2}$ h$^{-1}$ bar$^{-1}$.

Although there are various non-conventional adsorbent materials developed from hydrogels and biopolymers and used in the application of wastewater treatment, only a few studies were reported in the literature in this regard using real samples of PRWW. Moreover, most of the reported studies were conducted on a small laboratory scale. Hence,

there is a need for further research to understand the suitability of these materials in practical applications.

### 3.4.2. Electrochemical Techniques

Electrochemical technology is another promising treatment technology for the removal of organic pollutants from industrial wastewater using the application of electric currents supplied to electrodes. The major electrochemical techniques (Figure 12) can occur in the forms of electrocoagulation, electro-floatation electro-oxidation, electro-Fenton electrodialysis, electrodeposition, and electrode ionization etc. [136,137]. Meanwhile, this technique also has its major challenges and limitations which affect its friendly application and efficiency. Adetunji and Olaniran [9] reported that electrochemical technologies are usually affected by operating conditions such as current density, pH, electrode materials, temperature, concentration and structure of phenols. However, no chemical addition is needed and there is less waste generation, the electrochemical process is considered a green technology which is simple to operate and integrate with other techniques [137]. The average estimated time for an electrochemical treatment process to treat PRWW is about 5–6 h. However, most of the studies conducted on electrochemical treatments were lab-scale processes with only a few evaluated at a pilot scale. Hence, there is a need for further efforts to determine the applicability of the prototype technology of the system to establish its viability [138]. Furthermore, there is no universally accepted electrochemical treatment technology for the treatment of highly contaminated PRWW, but hybrid application with other treatment processes (such as biological and physicochemical processes) may provide the required efficiency and more work is needed in this direction [136,138].

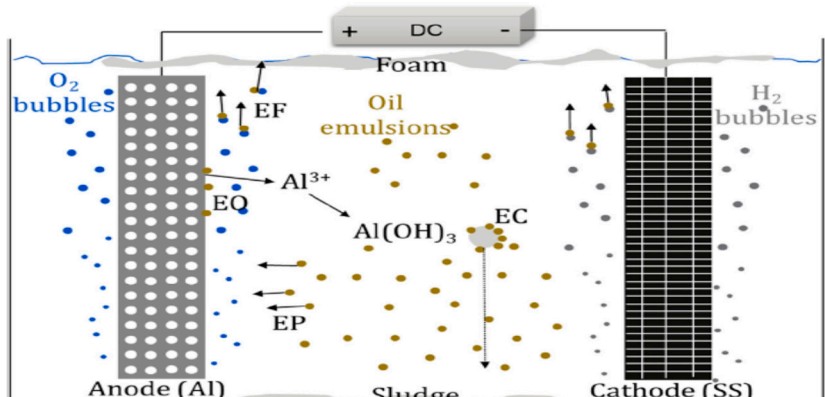

**Figure 12.** Electrochemical cells showing the different electrochemical processes [137]. DC: Direct current. EF: Electro-flotation. EC: Electrocoagulation EO: Electrooxidation. A: Aluminum. EP: Electrophoresis.

Electro Floatation (EF)

This is an advanced and enhanced form of air floatation process which carries floating contaminants to the surface by buoyancy and gas bubbles (usually oxygen and hydrogen gases) produced because of the electrolysis of water [137]. Unlike the conventional DAF which depends on the solubility of oxygen and nitrogen in the wastewater, in electro floatation oxygen and hydrogen gas bubbles are formed at the surface of the anode and cathode, respectively. Although it can be used as a separate process, it is usually combined with coagulation, flocculation, or both to remove contaminants by skimming [139]. The efficiency of the EF process is dependent on the current density, pH of solution and temperature. Furthermore, it differs from conventional air floatation in that it provides uniform and finely dispersed gas bubbles and requires little space and operational cost [9]. While the choice of the electrode material is vital to a successful implementation of EF, titanium-based inert anodes in the form of dimensional stable anodes (DAS) are the most dominantly used anodes [140]. Alam and Shang [141] studied the treatment of synthetic oil sand tailings using a batch cell electro-flotation reactor made up of a stainless-steel mesh

cathode and a Ti-IrO$_2$ mesh anode. At an optimum current density of 150 A/m$^2$, about 90% oil flotation efficiency was achieved.

Electrocoagulation (EC)

Electrocoagulation is one of the most prominent electrochemical processes employed in the removal of colloidal immiscible forms of pollutants of less than 10 μm [122]. This technology as indicated in Figure 13 involves an in-situ release of appropriate coagulant (such as aluminium or iron species) from a metal electrode with the application of an electric current. This process would lead to the electrolytic dissolution of the metal ions and result in a simultaneous formation of hydroxyl ions and hydrogen gas, while the coagulant aggregates and precipitates suspended solids [9]. Among the advantages of this technology are simple and automated operation, lower sludge volume and no chemical requirement, except for pH control [142]. Similarly, Akkaya [143] reported the use of aluminium and iron cathode electrodes obtained from scrap metals disposed of from industrial operations in the electrocoagulation process of PRWW under an optimum condition of 6.30 pH, current density of 22 mA/cm$^2$ and exposure time of 39 min. The process obtained COD and phenol removal efficiencies of 91.18% and 91.46%, respectively. However, a three-step pilot plant process investigated by El-Naas et al. [39] consisting of an electro-coagulation unit has resulted in the best performance to enhance COD and suspended solids removal. The plant achieved a 96% reduction in COD and a 100% reduction in phenol as well as cresol concentrations. El-Ashtoukhy et al. [144], also utilized a fixed-bed electrochemical reactor for the electrocoagulation of phenolic compounds in a real PRWW sample. They reported a 100% phenol removal efficiency of 3 mg/L in two hours. Gousmi et al. [145], similarly reported the application of iron and aluminium electrodes in an electrolytic reactor to determine the removal efficiencies of COD and turbidity from a synthetic PRWW. The process revealed 83.52% and 99.94% removal efficiencies for COD and turbidity, respectively. Meanwhile, the major drawback of this system, especially when used separately, often yields a lower efficiency in a high concentration of oily wastewater. Thus, it is commonly combined as an integrated process with other suitable techniques. Furthermore, it involves the application of electrochemical cells, where electrodes are dipped into oily wastewater, and determine the difference in the potential current being applied [5].

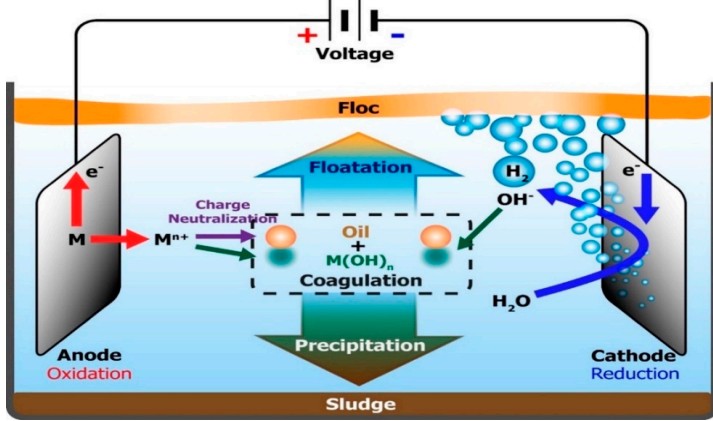

**Figure 13.** Mechanism of electrocoagulation process for oil removal [146].

Electrooxidation (EO)

This is an advanced form of chemical oxidation technique which involves the generation of the oxidants that oxidize the pollutants through the application of electric current [9]. The EO process is sometimes considered as part of the AOPs from a broad perspective but only the oxidation process occurs on the surface of the anode electrode as opposed to direct oxidation in the latter [137]. The efficiency of the EO technique is affected by operating conditions such as the current density and electrode activity as well as the pollutants' diffusion rate [9]. Ibrahim et al. [38] reported an electrochemical oxidation process for the

treatment of RWW effluent with optimized conditions of 30 mA/cm$^2$ current density, pH 8, supporting electrolyte 2 g/L, and operation time of 120 min. Ruthenium oxide-coated titanium and stainless steel served as the anode and cathode, respectively. FTIR analysis was conducted to determine the removal of pollutants by electrooxidation degradation, and an estimated 92% COD removal efficiency was obtained. The efficacy of lead oxide reinforced on tantalum (Ta/PbO$_2$) and boron-doped diamond (BDD) anodes contained in an electrolytic batch cell was determined by Gargouri et al. [147] for the treatment of oily wastewater. At different current densities of 30, 50 and 100 mA/cm$^2$ COD removal efficiency of 85% and 96% was obtained after 11 and 7 h, respectively. Some reported literature on the use of electrochemical techniques for the treatment of PRWW has been presented in Table 8.

**Table 8.** Treatment of petroleum refinery wastewater by electrochemical processes.

| Electrodes/Reactor | Process | Removal Efficiency | | | Reference |
| --- | --- | --- | --- | --- | --- |
| | | COD (%) | Phenols (%) | Oil (%) | |
| Porous graphite electrodes. | EFen | 95.9 | - | | [133] |
| Electrochemical reactor with Ti-IrO$_2$ mesh anode | EF | - | - | 90 | [141] |
| Aluminum and iron cathode electrodes from scrap metals | EC | 91.18 | 91.46 | - | [143] |
| Fixed-bed electrochemical reactor | EC | - | 100 | - | [144] |
| Aluminium electrodes in an electrolytic reactor | EC | 83.5 | - | - | [145] |
| Ruthenium oxide-coated titanium and stainless steel | EO | 92 | - | - | [38] |
| Lead oxide reinforced on tantalum (Ta/PbO$_2$) and boron-doped diamond (BDD) anodes | EO | 96 | - | - | [147] |

### 3.4.3. Advanced Oxidation Processes

Chemical oxidation techniques are a set of treatment processes which can be broadly classified into two types: conventional chemical treatments and advanced oxidation processes [5]. Advanced oxidation processes are highly efficient techniques used in the treatment of different types of wastewater including petroleum industry wastewater, toxic effluents from pharmaceutical industry wastewaters, etc. In previous years, several works have been reported in the literature to examine the efficiency of these processes [5]. Precisely, advanced oxidation processes (AOPs) are a category of chemical treatment methods that produce free hydroxyl radical groups with strong oxidant potential which are capable of degrading contaminants. The most commonly employed AOPs in the treatment of PRWW include Fenton and photo-Fenton oxidation reaction processes, electrochemical oxidation, ozonation processes (O$_3$), as well as heterogeneous photocatalytic oxidation [5,137]. The AOPs are gaining more attention nowadays as they are environmentally friendly techniques with less generation of secondary by-products and have shown high treatment efficiencies in the removal of organic compounds even at low concentrations [27]. The treatment capability is attributed to the strong hydroxyl radical ($^-$OH) which has strong reactivity towards organic compounds and colour degradation potential [134]. Based on this, AOPs have been reported as an efficient treatment technology for the reduction of COD, odour, colour, and other specific pollutants as well as sludge treatment. It can also be used in integration with biological treatment processes as a non-selective integrated chemical oxidant with high efficiency in removing toxic organic compounds such as phenols. Wang et al. [148] also reported that AOPs are usually rapid processes with high treatment efficiencies and little residual production, but on the other hand, associated with high energy requirements. Azizah and Widiasa [149] investigated the application of H$_2$O$_2$/UV and H$_2$O$_2$/UV/O$_3$ configurations for the treatment of PRWW with high phenol concentrations. High phenol degradation of about 93.75% was achieved using a H$_2$O$_2$/UV/O$_3$ configuration with 1000 ppm of H$_2$O$_2$ after 120 min. Several studies have also shown more than 90% COD and phenol removal efficiencies from the application of the H$_2$O$_2$-based advanced oxidation process. Similarly, de Oliveira et al. [150] synthesized TiO$_2$ nanoparticles assisted by microwaves, from titanium tetrachloride and water, and used it as a catalyst

subjected to photodegradation under UV irradiation using promising UF-Permeate from a membrane bioreactor. TOC and total nitrogen (TN) removal efficiencies were 32% and 67%, respectively, at a pH of 10 and catalyst concentration of 100 mg/L in a reaction time of 90 min. Furthermore, the catalyst showed stability after four different cycles of application and the data obtained was promising to improve the efficiency of the catalyst in the removal of organics. Most of the reviews such as [5,7,9,137,151] have shown the best experimental treatment results obtained using the AOPs rather than conventional methods.

Fenton-Oxidation

Generally, among the AOP techniques, the Fenton technology is found to be very attractive due to its simplicity, high performance, low cost as well and lack of toxicity of the Fenton reagents which are usually ferrous ion and hydrogen peroxide [152]. The Fenton process (FP) is based on a redox reaction between a chemical mixture of hydrogen peroxide $(H_2O_2)$ and ferric ions $Fe^{2+}$ which have a strong oxidizing potential in an acid medium. It is a technique which was founded by Henry John Horstman Fenton in 1894 [136]. The hydroxyl radical is capable of the degradation of toxic and non-biodegradable pollutants by direct or indirect anodic oxidation [9]. The OH radicals are extremely strong reactive oxidizers with an oxidation potential of approximately, $E\theta = 2.8$ V and they are generally non-selective towards organic pollutants in wastewater [152]. There are two types of Fenton reactions: the standard Fenton reaction which is formed by a reaction between ferrous iron $(Fe^{+2})$ ions and hydrogen peroxide $(H_2O_2)$, as well as the Fenton-like reaction which is formed by a reaction of $(Fe^{+3})$ ions and hydrogen peroxide [153]. The Fenton reaction conducted under intense light such as UV or sunlight which generates more hydroxyl radicals is called the photo-Fenton reaction. Normally the ratio between the iron ions and the peroxide which is $[Fe^{2+}]/[H_2O_2]$ is 1:2. However, the study reported by Quang et al. [154] suggested a ratio of 1:5 for a greater rate of degradation. The Fenton-oxidation technique has been widely investigated in the treatment of different types of wastewater effluents including textiles [154] and pharmaceuticals [155]. However, the Fenton process's general limitations include the problem of adding $H_2O_2$ and its lower utilization and mineralization efficiencies [155]. The review reported by Elmobarak et al. [5] also summarized that the major drawbacks of the Fenton and the photo-Fenton processes include their requirement for a very low pH value of usually less than 2 as well as the need for the elimination of the iron ions after the reaction process. Additionally, the potentiality of the $^-OH$ radical degradation tends to reduce with a rise in the pH value. At a very low pH, there would be a creation of Fe (II) $(H2O)^{2+}$ which can react with the hydrogen peroxide $(H_2O_2)$ leading to reduced generation of hydroxyl radicals. Shokria et al. [156] have reported a study using $FeCl_3$ and hydrogen peroxide on Box–Behnken design to decrease the COD of petrochemical wastewater. At a pH value of 5.63, a maximum COD removal efficiency of 72.06% and 74.9% were obtained at an operation condition of $[Fe^{3+}] = 1.76$ mM and $[H_2O_2] = 17.86$ mM. Other contaminants including the BOD, TOC and TDS also decreased considerably. Similarly, Tony et al. [157] have investigated the efficiency of using Fenton's reagent $(Fe^{2+}/H_2O_2)$ and photo-Fenton's reagent $(Fe^{2+}/H_2O_2/UV)$ for the treatment of oily PRWW from Whitegate refinery, County Cork, Ireland. At an optimized condition of pH 3, $H_2O_2$ (400 mg/L) and $Fe^{2+}$ (40 mg/L), the photo-Fenton treatment achieved approximately 50% COD removal efficiency. Similarly, the report from a study conducted by Hassan et al. [77], using the Fenton reagent $(Fe^{2+}/H_2O_2)$ also achieved 86% and 97% COD and total petroleum hydrocarbon (TPH) removal efficiencies at a pH of 3.5 and reaction time of 60 min. A heterogeneous Fenton-like degradation technique of organic pollutants from PRWW by copper-type layered double hydroxides to degrade aromatic and aliphatic organic compounds was reported by Radji et al. [158]. From their study, as indicated in Figure 14, they synthesized $Ni_{(2-x)}Cu_{(x)}Al$-LDH layered ternary double hydroxides as a catalyst with a series of x ratios: 0.0; 0.5; 1.5; and 2.0. Their findings on the oxidation reaction showed that catalytic activity varied inversely with the $Ni^{2+}/Cu^{2+}$ ratio and activity was maximum for x:2.0 where the catalyst can remove about 74.8% of TOC, and the aromatic

compounds. Hence, they conclude that $Cu^+$ is catalytically active and increases the TOC reduction in this case.

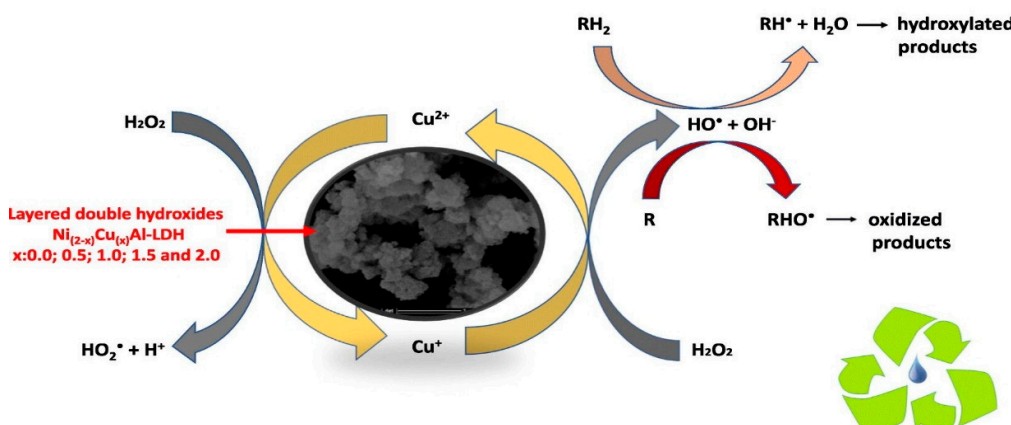

**Figure 14.** Heterogeneous Fenton-like degradation of PRWW pollutants using synthesized $Ni_{(2-x)}Cu_{(x)}Al$-LDH layered ternary double hydroxide catalyst [158].

Electro-Fenton Process

This is another novel oxidation technique which employs the electrochemical process and generation of $TiO_2$ oxidants by the Fenton-oxidation process. This technique has been studied for the removal of COD, BOD, TPH, phenols and other recalcitrant compounds which are not easily degraded in conventional treatment plants [6]. For example, Fahim and Abbar [151] have reported a study treating Al-Dewaniya PRWW in Iraq by the electro-Fenton process using porous graphite electrodes as anode and cathode materials. They used a tubular type of electrochemical reactor with a cylindrical cathode made from porous graphite and a concentric porous graphite rod which acts as an anode. At a current density of 25 A/cm$^2$, and operation time of 45 min with no addition of NaCl, the removal efficiency of COD was found to be 95.9% with an energy consumption of 8.921 kWh/kg per COD. The outcome of the experimental work has demonstrated the capability of the graphite–graphite electro-Fenton system as an effective technique in the removal of COD from petroleum wastewater. Similarly, Divyapriya and Nidheesh [159] also reported from their review that the use of graphene-based electrodes in the electro-Fenton technique is usually considered to be a promising and cleaner method to produce the reactive oxygen species that can rapidly mineralize organic contaminants. They also added that due to its catalytic activity, stability, and reusability, graphene derivatives have been used to immobilize various heterogeneous Fenton catalysts. On the other hand, the application of photovoltaic cell electro-Fenton oxidation has been reported by Atiyah et al. [160]. During the treatment process, hydrogen peroxide dosage, the electrolysis time, and current density including the rate of energy consumption and cost were examined for efficiency in the removal of TOC. The optimum operational conditions include current (0.5–2 mA), $H_2O_2$ concentration (10–50 ppm) and electrolysis time (10–30 min). Under these conditions, about 98% removal efficiency of the organic content was achieved with 39.67 kWh/m$^3$ of energy. However, only a few studies related to the use of the electro-Fenton process in the treatment of real PRWW have been reported. Synthetic wastewater prepared from demineralized water and phenol (0.5 g/L) has been used for the removal of the phenol in a study reported by Procházka et al. [161]. In their experiment, they used iron sulfate dosed into the solution (m = 0.261 g) to act as the source of $Fe^{2+}$ ions which constitute the iron anode, and the cathode is made of titanium to electrochemically enhance the reaction. A pH of 3 and a current density of 408.16 A/m$^2$ were employed. Subsequently, the dosed hydrogen peroxide provided free hydroxyl radicals and started the reaction with concurrently added iron ions. This process shows an excellent performance in the reduction of the COD until the postponement of the hydrogen peroxide addition. The major advantage of this process is the indirect addition of the $Fe^{+2}$ in the solid phase which eliminates the use of the Fe in

the form of its solution such as iron sulfate or iron chloride salts. However, besides the requirement for the addition of hydrogen peroxide catalyst, the major drawback associated with the electro-Fenton process which can prevent its industrial application is its energy consumption to support the electrochemical process.

Photocatalysis

Photocatalysis is nowadays regarded as one of the most advanced, as well as environmentally friendly techniques for the total degradation of organic contaminants in various forms of wastewater [162]. The term photo-catalysis as shown in the summary of processes in Figure 15 is a powerful chemical technology process which converts solar energy to chemical energy for the synthesis of highly functionalized complex molecules in the form of radicals [163–165]. Meanwhile, a photo-catalyst is often defined as a material such as titanium oxide ($TiO_2$) or transition metal oxides which can decompose harmful substances under the effect of sunlight containing UV rays [166]. The process occurs by the excitation of pairs of electrons in the valence band by UV which causes them to absorb higher energy than their band gap energy which then causes the simultaneous production of a hole in the valence band ($h^+$) and an electron ($e^-$) in the conduction band. Furthermore, the ($h^+$) and ($e^-$) species will then react with oxygen or water molecules to produce peroxide or hydroxyl radicals which are capable of degrading or decomposing organic compounds [27,166,167]. Depending on the specific characteristics of the semiconductor, the photolytic activity of photocatalysis is firstly initiated with the absorption of the energy in the form of photons which has an energy equal to or more than the band gap exhibited by the semiconductor such as the $TiO_2$ [168]. The process works on the basis that the hole created on the catalysts would in turn generate highly reactive hydroxyl radicals with high reduction–oxidation potentials such as $\bullet O_{2-}$, $H_2O_2$, and $\bullet O_2$ that can play an important role in the photocatalytic reaction mechanism [169]. Photocatalysis has been employed as a more advanced practical and efficient process in the treatment of wastewater to degrade organic contaminants [152,167,169]. In achieving this, pore volume, pore structure, crystalline sizes, light intensity as well as specific surface area are the important parameters which determine the excellent performance of photocatalysts. Abeish, [168] further noted that important operational parameters affecting the degradation of organic pollutants from PRWW using photocatalysis include temperature, pH, photocatalyst loading, wavelength and light intensity, initial pollutant concentration as well as TOC and COD concentrations. For example, a laboratory study reported by Pardeshi and Patil [170] revealed that the degradation of phenol is more effective under solar light intensity than artificial visible light irradiation. Furthermore, in the photocatalytic degradation of phenols and their chlorophenol and nitrophenol derivatives, hydroxyl radicals usually attack cyclic carbon atoms at the main reaction site, leading to the formation of various oxidation intermediates. The intermediates are then eventually converted to acetylene, maleic acid, carbon monoxide and carbon dioxide [171]. The $CO_2$ produced during the photocatalytic process can be trapped for other uses to prevent further environmental pollution [30].

Catalysts are the key requirement in the photocatalytic technique. A nano-catalyst usually possesses high surface area and density which gives it more photocatalytic activity and applicability in wastewater treatment [172,173]. For example, a titanium dioxide ($TiO_2$)-based photocatalyst is the most widely used in wastewater treatment due to its high oxidizing ability of organic compounds, cost-effectiveness, nontoxicity, and environmental friendliness [167,174,175]. Graphene, which is a carbon-based material has also been tested for photocatalysis applications and demonstrated high potential applicability for general pollutant removal. Dang et al. [176] reported that the most important semiconductor catalyst widely employed for photocatalytic degradation of phenols includes: ZnO, CdS, $TiO_2$, GaP, ZnS and $Fe_2O_3$. Meanwhile, Park et al. [169] also reported that the most widely studied and developed pollutant removal photocatalysts are titanium dioxide ($TiO_2$) and transition metal oxides. Although photocatalysis is an efficient technique, especially for the removal of organic pollutants, fast recombination of charge carriers is one of the major

limitations in the photocatalytic performance associated with many photocatalysts such as $TiO_2$ [177]. The UV light interaction with a photocatalyst works within a wavelength range of 280–400 nm and 400–700 nm for the visible light range [154]. It is important to note that the UV light intensity and initial concentration are very influential factors which affect the performance of photocatalytic degradation. Moreover, the efficiency of photocatalysis can be enhanced through the combination of photocatalysts with oxidizing agents such as $H_2O_2$ [178].

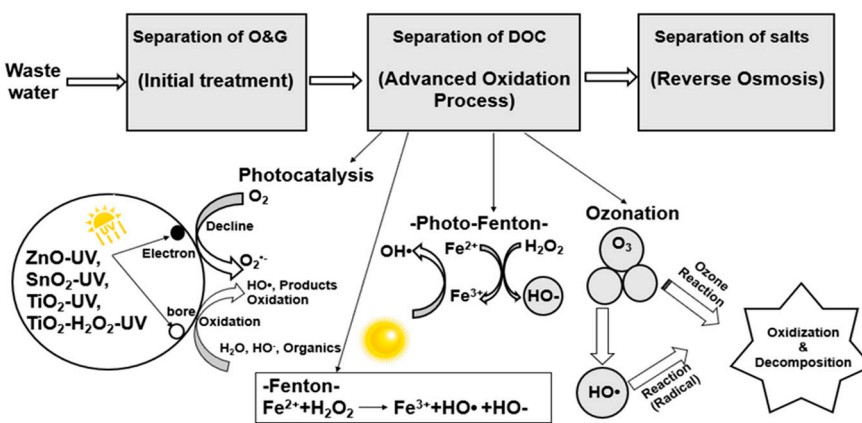

**Figure 15.** Summary of the advanced oxidation processes in the treatment of petroleum wastewater [5].

Properties of the Photocatalysts

Photocatalysts are usually employed either in the form of powders or thin films based on the requirements and scope of their application [179]. The nano forms of photocatalysts are better at fast reaction rates than their corresponding bulk forms due to their small size and high surface area [180]. However, using nanoparticles for wastewater treatment and pollutant degradation also has limitations related to their fast recombination losses and inadequate solar spectrum utilization [181]. For example, Estrada-Flores et al. [182] reported a study on the relationship between morphology, porosity, and the photocatalytic activity of the anatase phase of $TiO_2$ synthesized by a modified sol–gel using a different ionic surfactant. The results of the experiments show that the specific surface area of the anatase photocatalyst increases with an increase in pore sizes and that sodium dodecyl sulfate (SDS) modified anatase has the lowest band gap value of 2.97 eV and highest specific surface area of 138.72 $m^2$/g as well as the highest photocatalytic activity. Other important photocatalysts reported to have been used in the treatment of PRWW and their corresponding band gaps include $Fe_2O_3$ (2.2 eV), $\alpha$-$FeO_3$ (3.1 eV), $SnO_2$ (3.5 eV), Si (1.1 eV), $SrTiO_3$ (3.4 eV), ZnS (3.2 eV) and $WO_3$ (2.7 eV).

Metal Doping and Hybridization of Photocatalysts

Catalytic oxidation tends to increase with doping and hybridization modification processes, which increases the hydrophobicity and pollutant absorption strength of a photocatalyst [169]. To increase or intensify the photocatalytic activity of a catalyst, doping techniques are applied to improve sensitivity to UV light as well as reduce the band gap and recombination rate [167]. Besides improving photocatalytic activity, doping of catalysts also tends to reduce the amount of energy and wavelength required to be absorbed [167,169]. Metals such as iron [183], zinc [184], silver [185], platinum [186], or non-metal elements such as nitrogen [187] carbon and sulfur [188] have been employed as metallic dopants to enhance photocatalytic performance. On the other hand, catalyst hybridization is another technique used to enhance the degradation potential of organic pollutants where photocatalysts are sometimes combined with absorbents such as graphite, $SiO_2$ and hydroxyapatite [189]. A study conducted by Barakat et al. [190] on photocatalytic oxidation using an $H_2O_2$/UV/$TiO_2$ hybrid system for the degradation of phenols and mono chlorophenols (CP) revealed that combination of $TiO_2$ and $H_2O_2$ under UV illumination greatly enhanced

the degradation rates of the contaminants from 30% to 97% efficiency. The study further reported that more hydroxyl radicals are produced with the increase in the concentration of hydrogen peroxide ($H_2O_2$) and subsequently higher phenol oxidation rate.

Treatment of PRWW Using Photocatalysis

Due to the potential application using sunlight as a source of energy, this makes it an attractive technique for the degradation of organics from PRWW [172]. The degradation of organic contaminants from PRWW by photocatalysis (Table 9) has been widely investigated using various forms of photocatalyst under varying conditions. In this work, we review the recent studies conducted in the photocatalytic removal of organic contaminants and hope to provide prospects for the development of complex structured photocatalysts. Ghasemi et al. [191] reported the treatment of PRWW by photocatalytic degradation using the $TiO_2$/Fe-ZSM-5 photocatalyst with as-synthesized Fe-ZSM-5 zeolite produced via sol–gel method with a specific surface area of 304.6 $m^2$/g and 29.28% loaded $TiO_2$. About 80% COD removal was achieved at a pH level of 4, photocatalyst concentration of 2.1 g/L, and 45 °C UV exposure temperature through 240 min. Although high COD removal efficiency was achieved, the synthesis of ZSM-5 zeolite catalyst is associated with a high production cost via complex processes influenced by the effect of time and temperature [192]. The study reported by Ul haq et al. [35] using $Fe_2O_3$, $MnO_2$, $TiO_2$ and ZnO for the photocatalytic oxidative degradation of hydrocarbon pollutants from PRWW of Attock oil refinery in Pakistan showed that the highest photocatalytic degradation was exhibited by $TiO_2$ converting benzene, toluene, phenol, and naphthalene at 92, 98.8, 91.5, and 93%, respectively. The reaction conditions include a 100 mg/L catalyst dose at a pH of 3 and a temperature of 30 °C through a 90-min reaction contact time. Moreover, 93.2% COD removal efficiency is also recorded. Complete photodegradation of the parent hydrocarbons is observed using the UV/$TiO_2$ system. This study has proved a reference for the photocatalytic degradation ability of the $TiO_2$ photocatalyst and its low pH requirement and high COD efficiency removal.

Similarly, Aljuboury et al. [193] have reported the results of an investigation using combined solar photocatalyst titanium oxide/zinc oxide ($TiO_2$/ZnO) with aeration processes. The maximum removal efficiencies for TOC and COD removal were achieved at 99.3% and 76%, respectively. The optimum operating conditions included: 0.5 g/L each for $TiO_2$ and ZnO dosage, pH 6.8, air flow of 4.3 L/min and reaction time of 170 min. Fernandes et al. [194] have reported the synergistic effects of using $O_3$, $H_2O_2$ and $O_3$/$H_2O_2$ as an external oxidant with $TiO_2$ under intense UV light and evaluated the reduction in COD and BOD as well as degradation of Volatile Organic Carbons (VOCs). After 280 min of the treatment process, 38 and 32% COD and BOD reduction was observed and up to 84% degradation of the total VOCs. Furthermore, sulfide ion concentration was completely depleted in the first 30 min of the experiment. However, for an industrial application, there is a need to scale up a pilot scale of the process to a real-case scenario. Furthermore, titanium oxide and a silver doped nanoparticle synthesized as a $TiO_2$/Ag photocatalyst fixed on lightweight concrete plates were used in a study reported by Delnavaz and Bos'hagh, [195] on real PRWW. Determination of the effects of pH and mass loading on the system efficiency showed that the highest removal efficiency was in the pH range of 4.5 and a dosage of 15 g/$m^2$, respectively. The COD removal efficiencies recorded after 8 h under sunlight and using a UV-A lamp for $TiO_2$/Ag photocatalysts were 51.8% and 76.3%, respectively. Based on these experiments, the synthetic $TiO_2$/Ag photocatalysts are capable of the removal of COD from real PRWW both under sunlight and UV light intensities.

Photocatalytic efficiencies of $TiO_2$ and zeolite for the removal of COD and $SO^{2-}$ from PRWW were compared using a photocatalytic system by Tetteh et al. [196]. The operating conditions of the system include a catalyst dosage of 0.5–1.5 g/L, a mixing rate of 30–90 rpm and an 18 W UV light. After a reaction time of 15–45 min, the results show almost the same efficiency of 92% for zeolite and 91% for $TiO_2$. Similarly, oil removal efficiency by photocatalysis has been examined by Mohammed et al. [197] using a ZnO

photocatalyst under a solar light to determine the effects of dosage, pH and initial oil concentration. The outcome of the experiment shows 75% oil reduction, and the optimum catalyst concentration was found with a 3 g/L dosage of ZnO and a pH of 10. The oil degradation rate decreased with increasing oil concentration which might be due to an increased level of turbidity because of the oil suspension which in turn decreased the permeation of the solar light intensity. Hence, based on this, it can be ascertained that zinc oxide catalyst is very efficient at a high pH level. Similarly, a phenol degradation capacity of ZnO nanorods (NRs) grown on a glass substrate has been investigated by Daher et al. [198] who reported about 90% phenolic degradation under 254 nm UV light energy within 10 h of irradiation time. Local South African oil refinery wastewater was also treated by Tetteh et al. [199] via photocatalytic degradation using $TiO_2$ Degussa P25 catalyst comprising 80% anatase, and 20% rutile. Similarly, a batch-aerated photocatalytic reactor was used at different levels of operational variables including $TiO_2$ dosage (2–8 g/L), reaction time (30–90 min), as well as airflow (0.768–1.48 L/min). The optimum condition for phenol removal up to 76% was achieved at 8 g/L $TiO_2$ dosage, 90 min reaction time and 1.225 L/min aeration flow rate. Additionally, the photocatalytic degradation of phenols experiment conducted by Ramachandran et al. [200] using 0.2 g/L of $TiO_2$ as photocatalyst and employing 8 W UV lamps revealed that COD concentration is completely reduced after 5 h of reaction time. However, despite the efficiency of the UV lamps, they also reported that solar-supported photocatalysis is better considering the implications of time, space and cost. Aljuboury and Shaik [201] have reported a study conducted to investigate COD and TOC removal efficiencies from real PRWW using a hybrid combination of $TiO_2$/ZnO/air/Solar and $TiO_2$/ZnO/Fenton/Solar processes. About 74% of COD and 99% of TOC removal efficiencies were achieved after 180 min under an optimal condition of 54 g/L and 50 g/L ZnO and $TiO_2$ dosages, respectively. The report further revealed that the solar photocatalyst of $TiO_2$/ZnO/Fenton is most efficient at neutral pH and hence no pH level adjustment during the treatment process is needed. Although the $TiO_2$/ZnO/air process is costly, it is found to be more efficient in the situation where the pH level is greater than 7. It can be deduced from this and the previously reported studies that acidic and alkaline conditions of the PRWW determination are significant in choosing an appropriate photocatalyst and its application. A review summary of the treatment of petroleum wastewater using different sets of photocatalysts with this technique is presented in Table 9.

3.4.4. Combined $H_2O_2$/UV Advanced Oxidation Processes

The use of hydrogen peroxide as an oxidant in combination with potential sources of photon energy which can generate $HO^-$ radicals has been reported to be more successful in providing the hydroxyl radical that can degrade organic pollutants [151]. Using UV radiation of wavelengths >300 nm, $H_2O_2$ can decompose and generate $HO^-$ radicals [5]. Different researchers have reported that the degradation of pollutants by $H_2O_2$ continues steadily up to its highest efficiency after which it starts to decrease. This sudden decrease has been proven to be a result of the generating hydroxyl radicals which start to react with the additional $H_2O_2$ [202]. However, different advantages can be mentioned for the application of the $H_2O_2$/UV oxidation process, for example, there is no requirement for the removal of the hydroxyl radical after the treatment process and it is completely soluble in water [203]. Elmobarak et al. [5] reported that the optimum operating pH using the $H_2O_2$/UV oxidation process should usually be with a pH < 4 to avoid the impact of ionic radicals such as carbonate and bicarbonate ions.

**Table 9.** Treatment of PRWW using photocatalysis techniques.

| Kin Photocatalyst | Experimental Conditions | | | | | Efficiency (%) | | | Reference |
|---|---|---|---|---|---|---|---|---|---|
| | Light | pH | Dosage | Temp. (°C) | Time (Min) | COD | Phenols | Oil | |
| TiO$_2$/Fe-ZSM-5 | UV | 4 | 2.1 g/L | 45 | 240 | 80 | - | - | [191] |
| TiO$_2$ | UV | 3 | 100 mg/L | 30 °C | 90 | 93.1 | 98.8 | - | [34] |
| TiO$_2$/ZnO | Solar | 6.8 | 0.5 g/L | NR | 170 | 76 | - | - | [193] |
| TiO$_2$ with synergistic effects of O$_3$, H$_2$O$_2$ and O$_3$/H$_2$O$_2$ | UV | NR | | 280 | | 38 | - | - | [194] |
| TiO$_2$/Ag | Solar/UV | 4.5 | NR | NR | NR | 51.8/76.3 | -- | - | [195] |
| Zeolite and TiO$_2$. | UV | NR | 0.5–1.5 g/L | | | 92/91 | | -- | [196] |
| ZnO | Solar | 10 | 3 g/L | NR | NR | | -- | 75 | [197] |
| TiO$_2$ Degussa P25 (80% anatase and 20% rutile) | UV | NR | 8 g/L | | | | 76 | - | [199] |
| TiO$_2$ | UV | NR | 0.2 g/L | NR | 300 | 100 | - | - | [200] |
| TiO$_2$/ZnO | UV | 7 | 54 g/L and 50 g/L | | | 74 | - | - | [201] |
| ZnO nanorods (NRs) | UV | NR | NR | NR | 600 | | 90 | - | [204] |

*3.5. Integrated Treatment Processes (ITP)*

It can be noted that most conventional and advanced treatment techniques have specific limitations in terms of their efficiency and are sometimes associated with various demerits for the treatment of PRWW. Based on this there is always a key interest in developing a novel procedure that overcomes limitations such as the operational costs, treatment efficiency, and generation of secondary pollutants. These challenges can sometimes be addressed through an integrated or combined treatment process which can yield a better benefit. For example, the combination of AOP techniques integrated with conventional methods used for the treatment of different contaminated industrial wastewater has been confirmed to be more efficient. However, as indicated in Figure 16 the development of an integrated treatment process requires a good understanding of the PRWW characteristics, cost determination, as well as the requirements of environmental policies [5].

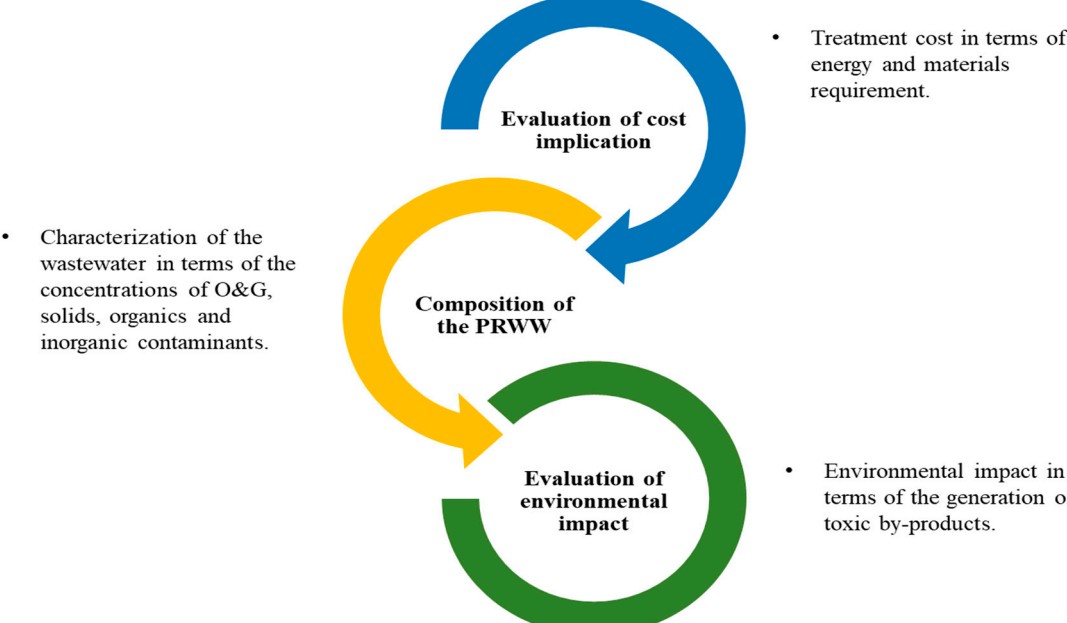

**Figure 16.** The basic considerations in the selection of an appropriate treatment technique.

In most cases, the development of an integrated treatment process is initiated from a combination of two or more conventional and advanced methods. Hence, the configurations of such a combination can be a two-step, three-step or even multiple-treatment process. This might comprise at least one conventional and one advanced, two conventional and one advanced, or even two advanced processes. For example, an advanced oxidation process integrated with biological or other treatment techniques increases the efficiency of degradation as well as the separation of the contaminants. Based on this, the biological processes can provide the needed decomposition of the residual oil and degradation etc. [205]. On the other hand, the integration of biological methods with membrane-based AOP techniques was also found to be an efficient process for the treatment of PRWW. However, in the development of a biological and chemical integrated process, the determination of the individual biological activity and chemical oxidation efficiencies is important for finding the optimal operating conditions for the combined process. This involves a profound knowledge of the operational conditions for both the biological and chemical processes. Hence, as indicated by Oller et al. [205] several analytical parameters, such as the COD and TOC concentration, must be monitored during each step of the treatment stage [205]. The study by Obuebite and Okwonna [206] reported for treating PRWW from Port Harcourt refining company in Nigeria employed an integrated system using a biological aerated lagoon and UV light degradation. With a pH of 7.84, the efficiency of the process yielded BOD 0.65 mg/L, COD 1.87 mg/L, TDS 69.96 mg/L and TSS 14.82 mg/L. Similarly, Ul Haq, et al. [35] carried out integrated photocatalytic oxidation and adsorption processes to treat PRWW using a $TiO_2$/Activated Carbon (AC) hybrid material. The hybrid adsorbent/catalyst was prepared by impregnating $TiO_2$ over an activated carbon. Under the optimized reaction conditions of pH 3, temperature 30 °C, and 1000 mg $TiO_2$/AC per 500 mL of the sample and a contact time of 50 min., the integrated photocatalytic oxidation-adsorption process achieved a net percentage removal of benzene, toluene, aniline, and naphthalene concentrations of 91% from model hydrocarbons (HCs) solutions. Applying the same process under the same conditions for the treatment of real samples using $TiO_2$ and AC caused a 95% decrease in (COD) but at a longer contact time of 105 and 90 min, respectively. The impact of taking longer contact time usually results in the use of higher adsorbent and catalyst doses. However, the integrated photocatalytic oxidation and adsorption techniques using the hybrid $TiO_2$/AC showed a better advantage over the individual adsorption and photocatalytic oxidation processes using an individual application of AC and $TiO_2$. The combination of a membrane and bioreactor has also been popular in the degradation of PRWW. Razavi and Miri [63] reported the treatment of real PRWW using a hollow fibre membrane bioreactor (HF-MBR). The bioreactor included an ultrafiltration membrane (UF) and the HF-MBR was run for 160 days. The results of the process indicated an excellent average elimination of the COD, $BOD_5$, TSS, volatile suspended solids (VSS) and turbidity at 82%, 89%, 98%, 99%, and 98%, respectively. The efficiency of the process was found to be excellent, but the problem of a long duration time for biological degradation is mostly a common limitation associated with biological processes using bioreactors. Furthermore, the study reported by El-Naas et al. [25] utilized a three-step integrated process consisting of an electrocoagulation cell (EC), a spouted bed bioreactor (SBBR) and an adsorption column. The reactor contains *Pseudonymous putida* immobilized in polyvinyl alcohol gel, while the adsorption column was packed with granular activated carbon adsorbent produced from agricultural waste. This process was able to achieve a reduction in COD, phenol, and cresol concentrations by 97%, 100% and 100%, respectively. They reported that the process can handle highly contaminated PRWW with a relatively wide range of operating conditions. Similarly, Wang et al. [57] have reported the treatment of PRWW using a multistage-enhanced biochemical process. The technique comprises different units of biological aerated filter (ICBAF), hydrolysis acidification (HA), two anaerobic–aerobic (A/O) units, a membrane biological reactor (MBR), and ozone-activated carbon ($O_3$-AC) units. Hence, the integrated system in this method is comprised of biological, chemical, and adsorption processes. They revealed that

the overall efficiency of the system has achieved 94% removal of COD, $BOD_5$, ammonia nitrogen ($NH^{4+}$-N) and phosphorus. The ICBAF unit accounts for 54.6% of COD removal and 83.6% of $BOD_5$ removal, and the two A/O units account for about 33.3% and 9.4% of the COD and $BOD_5$ removal efficiencies, respectively. Conventional biological systems such as biological aerated filters (BAF), and membrane bioreactors are usually environmentally friendly, however, they are often inefficient in removing total pollutants from PRWW that have high COD concentrations above 2000 mg/L. Keramati and Ayati [207] have reported from their study of the treatment of PRWW using a combination of electrocoagulation and photocatalytic processes. ZnO nanoparticles immobilized on a concrete surface were used as the photocatalyst to evaluate the efficiency of the system for COD removal. The system's efficiencies were first determined individually before integrating them and evaluating the optimum operating conditions. At a COD concentration of 900 m/L, the optimum condition of the EC process was 20 mA/cm$^2$ current density, 8.5 pH and 0.5 g/L NaCl concentration. Based on these conditions, a COD removal efficiency of about 94% was obtained after 60 min contact time. Meanwhile, for their experiment using the photocatalytic process, they used a COD concentration of 600 mg/L at optimum conditions of 80 g/m$^2$ ZnO concentration, pH of 5 and 32 W irradiation power. The COD removal efficiency was 76% after 300 min. Thereafter, they implemented the integrated EC and photocatalytic system using an initial COD concentration of 1000 mg/L where a COD removal efficiency of 47% was achieved after 8.5 min using the EC process. Finally, the effluent entered the concrete photoreactor for 120 min, which led to an 85% decrease in the COD concentration. Similarly, Ratman et al. [108] have conducted a study to explore the application of an advanced membrane process integrated with ozonation as a preliminary treatment. They used a polysulfone PSf-TiO$_2$ membrane and a constant ozonation dose of 3000 mg/h at different times and temperature combinations. They noticed a longer ozonation time significantly improved the removal of pollutants. However, an increase in temperature does not significantly affect COD, phenol and TDS, removal efficiencies with this system, but only ammonia removal up to 82%. The use of the ozonation process also enhanced the permeate flux of the membrane by up to 96% and improved pollutant removal efficiency by up to 77%. This integrated process might be a good option for the treatment of PRWW with high ammonia concentrations. On the other hand, Mokhtari et al. [208] have also reported the application of an innovative method of a biological process coupled with a sand filter column for the treatment of Iranian PRWW as a hybrid process. It is a simple integrated process where the sand filter column is used in the last part of the treatment process. The biological system consisted of four fully immersed vertical rotating bioreactors (RBCs) with the sand column filter placed at the end. Overall treatment efficiencies recorded for COD, TSS, oil, ammonia (NH$_3$), and turbidity were 94%, 90%, 88%, 93%, and 92%, respectively. These results have also confirmed the effectiveness of the integrated system in achieving high removal efficiencies in the treatment of PRWW.

## 4. Conclusions

Environmental pollution due to oil refinery wastewater is a global phenomenon that attracts serious attention due to its harmful effects on the ecosystem. This review has presented an overview of the recent application of conventional and advanced treatment techniques in this regard. Nowadays, the need to meet the maximum concentration limit for PRWW is usually challenging for petroleum refinery industries. This is because petroleum wastewater has a dynamic, complex nature. Hence, various conventional and advanced treatment techniques such as adsorption, membrane filtration, chemical precipitation, and biological systems have been designed to address this challenge over the years. While some already established techniques are mature in their applications, others are associated with various challenges and limitations. The appropriate treatment technology selection mostly depends on the oily wastewater composition, operational costs, efficiency, and environmental impacts. However, as the nature of PRWW is mainly in the form of oil-in-water emulsions, a correct understanding of their physical and chemical composition

is needed. Although membrane treatment techniques have demonstrated an efficient removal capacity for organic and inorganic contaminants, they are associated with fouling and the problem of salt build-in bioreactors. New and advanced treatment techniques such as adsorption with modified non-conventional adsorbents, photocatalysis, and other advanced oxidation processes have been reported with significant efficiency in refinery wastewater treatment. For example, photocatalysis treatment techniques are effective in reducing COD, oil and grease concentrations, and phenol degradation. The alternative use of solar power as an energy source also makes it a considerable treatment option. Meanwhile, adsorption techniques using non-conventional adsorbents such as hydrogel were also effective in treating synthetic petroleum wastewater. They are less costly as well as environmentally friendly in their application.

**Future research perspective**

Because the use of a single treatment technique does not usually yield the required total treatment efficiency, the application of an integrated or hybrid process is nowadays taking much interest in designing a novel procedure that overcomes various treatment limitations. An integrated treatment approach using conventional and advanced techniques would improve the efficiency of many treatment processes. Using modified adsorbents from biopolymers and hydrogels, natural and geomaterials, and agricultural by-products has received little work on treating real PRWW. Meanwhile, most reported studies used synthetic wastewater samples at the laboratory scale. The behaviour of the adsorbent might sometimes change due to different characterizations of a real sample. Based on this, treatment techniques must be tested using a real sample to ascertain their validity in application at the industrial level. This could perhaps lead to the development of innovative treatment techniques for the petroleum industry, promoting water sustainability. Furthermore, waste recovery to develop social and environmental sustainability is taking focus from the recovery of wastes produced by humans and various industrial sectors. As significant waste, including oil/grease and metals, is produced from the petroleum refinery industry, an effective recovery from these pollutants can also enhance environmental health and the application of resources.

**Author Contributions:** Conceptualization, M.S.L., R.K. and M.A.E.-F.B.; software, M.S.L.; validation, M.S.L., R.K. and M.A.E.-F.B.; investigation, M.S.L., R.K., J.R. and M.A.E.-F.B.; resources, R.K. and M.A.E.-F.B.; data curation, M.S.L.; writing—original draft preparation, M.S.L.; writing—review and editing, R.K., J.R. and M.A.E.-F.B.; supervision, R.K. and M.A.E.-F.B.; project administration, R.K. and M.A.E.-F.B.; funding acquisition, R.K. and M.A.E.-F.B. All authors have read and agreed to the published version of the manuscript.

**Funding:** This project was funded by the Deanship of Scientific Research at King Abdulaziz University, Jeddah, under grant no. G: 143-155-1441.

**Acknowledgments:** This project was funded by the Deanship of Scientific Research at King Abdulaziz University, Jeddah, under grant no. G: 143-155-1441. The authors, therefore, acknowledge with thanks DSR for technical and financial support.

**Conflicts of Interest:** The authors declare no conflict of interest.

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
