# Peer review of "Recent Advancements in the Treatment of Petroleum Refinery Wastewater"

_water, doi:10.3390/w15203676_

Round 1

Reviewer 1 Report

The manuscript titled "Recent advancement in the treatment of petroleum refinery wastewater" carried out a deepen analysis the treatment of petroleum refinery wastewater (PRWW) in the field of industrial wastewater management. In my opinion, the manuscript is interesting and appropriate for the present journal. I think that it can be accepted under major revisions.

1. There are many paragraphs in the article that quote the content of a literature, and there is a lack of relevant comments. It is suggested that the author add comments and integrate relevant paragraphs.

2. Table 4,5,6,7 in the paper are not standardized, and the number of citations is too low, it is recommended to add about 10 additional references per table.

3. The conclusion section is too general, and it is recommended to summarize each item by category.

4. Page no.29, authors are suggested to add a separate section of future perspective.

5. In the introduction section, the author suggests adding some references when describing the application of absorbent to water treatment. The following papers can be cited:

[1] Biochar as a low-cost adsorbent for aqueous heavy metal removal: A review. Journal of Analytical and Applied Pyrolysis, 2021, 155: 105081.

[2] Application of biochar for the adsorption of organic pollutants from wastewater: Modification strategies, mechanisms and challenges. Separation and Purification Technology, 2022, 300: 121925.

6. In the section summarizing the different treatment methods, it is suggested to add a detailed reaction mechanism diagram.

The use of English is not at the desired standards and needs great improvement. The manuscript is quite often incomprehensible.

Author Response

[Water] Manuscript ID: water-2601280 Responses and changes to the manuscript titled "Recent advancement in the treatment of petroleum refinery wastewater"

COMMENTS REVIEWER 1:

  • There are many paragraphs in the article that quote the content of the literature, and there is a lack of relevant comments. It is suggested that the author add comments and integrate relevant paragraphs.

Response

Relevant comments were integrated to explain the content of the literature. According to the author's guide, the highlights are included in the manuscript.

  • Tables 4,5,6,7 in the paper are not standardized, and the number of citations is too low, it is recommended to add about 10 additional references per table.

Response

More relevant citations were added to the Tables 4, 5, 6 and 7. The Tables were now updated to a standard according to the author's guide.

  • The conclusion section is too general, and it is recommended to summarize each item by category.

Response

The content of the conclusion section is now summarized based on each category of the items discussed.

  • Page no.29, authors are suggested to add a separate section of future perspective.

Response

A separate section titled “future perspective” is added to highlight the content of the future research prospects.

  • In the introduction section, the author suggests adding some references when describing the application of absorbent to water treatment.

Response

Some relevant reference citations describing the application of absorbent to water treatment were included in the introductory section

  • In the section summarizing the different treatment methods, it is suggested to add a detailed reaction mechanism diagram.

Response

A detailed reaction mechanism diagram where it is necessary relevant to describe the process was included in the sections summarizing different treatment methods. However, not all methods have gotten separate individual diagrams to minimize the pages of the manuscript.

  • The use of English is not at the desired standards and needs great improvement. The manuscript is quite often incomprehensible.

Response

The quality of the English language grammar has been improved and proofread carefully.

Thank you very much for your valuable comments.

Reviewer 2 Report

Dear Editor

Thank you for the invitation to review the review manuscript entitled "Recent advancement in the treatment of petroleum refinery wastewater". This review manuscript described the recent research studies applied in the treatment of petroleum refinery wastewater using conventional, advanced as well as integrated treatment techniques. Also, the authors present in the review the major limitations of each technique as well as prospects for improvement.

The current work is seems good but it is not possible to publish in Water because all Figures and Tables (except Figure 1 and Table 1) were not mentioned in the text of the review manuscript.

Comments to the authors:

Major comments:

1.      All Figures and Tables (except Figure 1 and Table 1) were not mentioned in the text of the review manuscript.

2.      The title of the manuscript is not comprehensive because the author explains both the conventional treatment methods and the advanced methods.

3.      Paragraph 2, Page 2, Fig 1 and Fig. 2 are out of context.

4.      In page 5 and 6 there are many repetitions of the type of pollutants in refinery wastewater.

5.      I suggest to add conclusion better than summary in the last section of the review.

It seems good English Language

Author Response

[Water] Manuscript ID: water-2601280 Responses and changes to the manuscript titled "Recent advancement in the treatment of petroleum refinery wastewater"

Reply to reviewer 2

COMMENTS REVIEWER 2:

  • All Figures and Tables (except Figure 1 and Table 1) were not mentioned in the text of the review manuscript.

Response

All figures and tables were duly mentioned appropriately in the text of the review manuscript. However, it should be noted that most of the Tables were presented as a summary of the text earlier presented describing the various treatment techniques reported in the literature.

  • The title of the manuscript is not comprehensive because the author explains both the conventional treatment methods and the advanced methods.

Response

The term “advancement” otherwise referred to as improvement or progress here in the title of the manuscript is targeted to comprise all the relevant reported conventional and advanced treatment methods. Hence, that is why we tried as much as possible to report only the most recent data in each treatment technique.

  • Paragraph 2, Page 2, Fig 1 and Fig. 2 are out of context.

Response

The purpose of Paragraph 2 on page 2 and Figures 1 and 2 express the comprehensive “methodology” used in the preparation of the review process. It tries to express the need in terms of the review literature on the context of the treatment of petroleum refinery wastewater. Hence, no discussion of any techniques is provided but only the methodology of the review to maintain a single point paragraph.

  • In page 5 and 6 there are many repetitions of the type of pollutants in refinery wastewater.

Response

The reputations of on the discussion of refinery wastewater pollutants were taken amended by merging the contexts of pages 5 and 6.

  • I suggest to add conclusion better than summary in the last section of the review.

Response

The Section “Summary” has been now changed to “Conclusion.”

  • Comments on the Quality of English Language. It seems good English Language.

Response

Thank you very much for your valuable comments and the recommendation on the Quality of English Language.

Reviewer 3 Report

Journal: Water

Title: Recent advancement in the treatment of petroleum refinery wastewater

This is a valuable review paper that reviews the physical, biological, chemical, and advanced methods of petroleum refinery wastewater treatment. In addition, the authors propose the concept of hybrid, i.e. integrated treatments with the aim of obtaining greater efficiency. The work is really voluminous and it takes time to study the entire text. However, there are many repetitions of the same topics in the work, such as the composition of petroleum waste water, especially in chapters 1 and 2. Therefore, chapter 1 and 2 should be thoroughly revised.

Within the sentence there are a lot of words written with a capital letter which is unnecessary, for example Sulfides.

Do not use unnecessary abbreviations, that is, when the abbreviation is mentioned for the first time, explain it in that place.

There are unnecessarily bold words and underlined words.

Write all chemical formulas correctly.

Write the measurement units correctly and separate them from the numerical values.

Uniform the abbreviations PRWW and RWW throughout the text. Use only one type of abbreviation.

The meaning of using Figures and Tables. They are just inserted into the text without explaining or referring to them in the text.

To strengthen the critical review of the exhaustively listed research results. Advantages, disadvantages in terms of performance, duration, challenges and the like.

SPECIFIC COMMENTS

Page 2:

The text in the second paragraph related to methodology is good, but in my opinion, it is in the wrong place.

Page 3:

In Figures 1 and 2, the ordinate is missing.

Figures 2 and 3, 4, 5, 6,7,8, 9, 10 are not mentioned in the text nether explained.

Chapter 2 has too much overlap with Chapter 1. Systematize.

Page 7.

Table 1 is mentioned in the text, but in reality there is no table 1.

Page 10:

Explain Figure 6. The letters in the picture are not legible. Meaning of abbreviations in the 2nd treatment.

Change the numbering of chapters 3.1.2-3.1.5. since they belong to chapter 3.1.1., physical-chemical processes. 3.1.2. change to 3.1.1.1. etc.

The first 7 lines of text in chapter 3.1.2. belongs to chapter 3.1.1....

In chapter 3.1.2, you mention filtration in the title, but in that chapter, filtration is neither mentioned nor explained.

Explanation of table 3....

Page 12:

In the first and second paragraphs of chapter 3.1.4. there is a lot of repetition.

„The sorption capacities reported are; 15.52 mg/g, 16.23 mg/g and 12.91 mg/g for the activated carbon, natural clay and sawdust respectively at 100 min. equilibrium time.“ - Define sorption capacity in relation to waste substances.

Page 13:

„80% activated carbon parking“ –parking ???

Refer to table 4 in the text and explain the table 4, 5, 6, 7, 8.

Chapter 3.1.5.: „Membrane technology has been in existence since around the 18 th century“ - Cite the literature for this information. Is the 18th century correct?

Page 14: Do not use subheading 3.2.1. or place the first two sentences of chapter 3.2.1 between chapter 3.2. and 3.2.1.

Page 15:

„High chlorine concentration usually favours the formation of hydroxo salts precipitates instead“ – chlorine or chloride?

Page 16:

„traditional activated sludge (ASP)“ - Is the abbreviation correct?

Page 21:

Chapter 3.4.3.: „In

previous years, several works have been reported in the literature been conducted to examine  the  effi ciency  of  the  advanced  oxidation  processes  in  the  treatment  of  diff erent wastewaters  containing  recalcitrant  and  toxic  pollutants  (Elmobarak  et  al.  2021).“ - Unusual sentence construction

Chapter 3.4.3.1.: In relation to the previous chapter, do not repeat the explanation related to the Fenton reagent.

Correctly write the OH radical, as HO with a dot and not a minus.

Page 23:

„At an optimized condition of pH 3, H 2 O 2  (400 mg/L) and Fe 2+  (40 mg/L), the photo-Fenton treatment achieved approximately 50% COD removal effi ciency.“ - Efficiency of removing what?.

Page 24: In chapter 3.4.3.2. briefly explain the method as in other chapters.

Page 26:

Chapter 3.4.3.2.: „ In other to increase“ or order?

Page 28: In the second paragraph, define the maximum removal of which substance.

In the third paragraph: SO2-, or SO42-?

Page 31: „900 m/L“ - Is it the correct measurement unit?

Grammatically incomplete sentences appear. The wrong words are used......

Author Response

[Water] Manuscript ID: water-2601280 Responses and changes to the manuscript titled "Recent advancement in the treatment of petroleum refinery wastewater"

Reply to reviewer 3

COMMENTS REVIEWER 3:

This is a valuable review paper that reviews the physical, biological, chemical, and advanced methods of petroleum refinery wastewater treatment. In addition, the authors propose the concept of hybrid, i.e. integrated treatments to obtain greater efficiency. The work is really voluminous and it takes time to study the entire text. However, there are many repetitions of the same topics in the work, such as the composition of petroleum wastewater, especially in chapters 1 and 2. Therefore, chapters 1 and 2 should be thoroughly revised.

Within the sentence, there are a lot of words written with a capital letter which is unnecessary, for example, Sulfides.

Do not use unnecessary abbreviations, that is, when the abbreviation is mentioned for the first time, explain it in that place.

There are unnecessarily bold words and underlined words.

Write all chemical formulas correctly.

Write the measurement units correctly and separate them from the numerical values.

Uniform the abbreviations PRWW and RWW throughout the text. Use only one type of abbreviation.

The meaning of using Figures and Tables. They are just inserted into the text without explaining or referring to them in the text.

To strengthen the critical review of the exhaustively listed research results. Advantages, disadvantages in terms of performance, duration, challenges and the like.

Response

Dear reviewer, thank you for your valuable comments.

Repetitions in the context of the manuscript have been addressed carefully with a thorough revision and amendments in chapters 1 and 2. Additionally, grammatical errors due to the spelling of small and capital letters have also been rectified. Abbreviations and use of chemical formulas were properly addressed with an inclusion of discussion on tables and figures. The manuscript's critical review has also been improved.

Specific Comments

Page 2:

The text in the second paragraph related to methodology is good, but in my opinion, it is in the wrong place.

Response

The methodology is just after the introduction paragraph before the main discussion. Hence, it is based on general article standards.

Page 3:

  • In Figures 1 and 2, the ordinate is missing.

Response

The ordinate is now shown in Figures 1 and 2.

  • Figures 2 and 3, 4, 5, 6,7,8, 9, and 10 are not mentioned in the text nor explained.

Response

      The content of the listed figures is now mentioned in the manuscript content.

  • Chapter 2 has too much overlap with Chapter 1. Systematize.

      Response

      The overlap in Chapters 1 and 2 has been addressed already.

Page 7:

  • Table 1 is mentioned in the text, but in reality, there is no table 1.

Response

The table is mentioned and it is the table showing the contaminants from PRWW including BOD, COD, TSS etc. It is not placed correctly due the preprints edition.

Page 10:

  • Explain Figure 6. The letters in the picture are not legible. Meaning of abbreviations in the 2nd treatment.

Response

The content of the figure is briefly mentioned in the previous text which provides a complete summary of the treatment techniques discussed in the subsequent texts. Additionally, the meaning of the abbreviations has been provided accordingly.

  • Change the numbering of chapters 3.1.2-3.1.5. since they belong to chapter 3.1.1., physical-chemical processes. 3.1.2. change to 3.1.1.1. etc.

Response

The Chapter numbering has been corrected.

  • The first 7 lines of text in chapter 3.1.2. belongs to chapter 3.1.1....

Response

The first 7 lines of text in chapter 3.1.2. have been moved to chapter 3.1.1....

  • In chapter 3.1.2, you mention filtration in the title, but in that chapter, filtration is neither mentioned nor explained.

Response

As filtration is not a single technique of its own but rather goes along with floatation and sedimentation, it is now removed from the heading.

  • Explanation of Table 3...

Response

The Table 3 contents are just a summary of the full previous text. Meanwhile, it has been also mentioned properly.

Page 12:

  • In the first and second paragraphs of chapter 3.1.4. there is a lot of repetition.

Response

The repetition from 3.1.4 has been corrected.

  • „The sorption capacities reported are; 15.52 mg/g, 16.23 mg/g and 12.91 mg/g for the activated carbon, natural clay and sawdust respectively at 100 min. equilibrium time. “- Define sorption capacity in relation to waste substances.

Response

The definition of the term sorption capacity has been cited properly.

  • Refer to table 4 in the text and explain the table 4, 5, 6, 7, 8.

Response

Tables were mentioned in the text.

  • Chapter 3.1.5.: „Membrane technology has been in existence since around the 18th century“- Cite the literature for this information. Is the 18th century correct?

Response

The Citation on membrane technology history is now provided.

Page 14:

  • Do not use subheading 3.2.1. or place the first two sentences of chapter 3.2.1 between chapter 3.2. and 3.2.1.

  Response

   The subheading and the sentence discrepancy are now corrected.

  • Page 15:

High chlorine concentration usually favors the formation of hydroxo salts precipitates instead “– of chlorine or chloride?

Response

The correct spelling of the word chloride is now corrected.

Page 21:

  • Chapter 3.4.3.: „In previous years, several works have been reported in the literature to examine the efficiency of the  advanced  oxidation  processes  in  the  treatment  of  different wastewaters  containing  recalcitrant  and  toxic  pollutants  (Elmobarak  et    2021). “ - Unusual sentence construction

Response

The unusual sentence on page 21 is now corrected.

  • Chapter 3.4.3.1.: In relation to the previous chapter, do not repeat the explanation related to the Fenton reagent.

Response

  • Fenton process description repetition is corrected.
  • Correctly write the OH radical, as HO with a dot and not a minus.

Response

The OH radical is now written correctly.

Page 23:

  • At an optimized condition of pH 3, H 2 O 2 (400 mg/L) and Fe 2+  (40 mg/L), the photo-Fenton treatment achieved approximately 50% COD removal efficiency.“ - Efficiency of removing what?.

Response

  • The content of the sentence in now corrected.

Page 24:

  • In chapter 3.4.3.2. briefly explain the method as in other chapters.

Response

Explanation is now provided.

Page 26:

  • Chapter 3.4.3.2.: „ In other to increase “or order?

Response

In order to is now correctly written as ‘in other to’.

Page 28

  • In the second paragraph, define the maximum removal of which substance. In the third paragraph: SO2-, or SO42-?

Response

The sentence is now corrected.

Page 31:

  • 900 m/L “- Is it the correct measurement unit?

Response

The milligram per litre unit is now properly written as m/l

  • Grammatically incomplete sentences appear. The wrong words are used...

Response

Overall, the English language grammar of the manuscript has been improved.

Round 2

Reviewer 1 Report

After amendments, the manuscript is now available in the journal.

some sentences  are need to be rewrited.

Reviewer 2 Report

Dear Editor

I would like to inform you that the authors answer the comments and therefore I recommend to accept in Water

Regards

Reviewer 3 Report

Since the authors accepted the suggestions and improved the manuscript compared to the previous version, my decision is to accept the manuscript in this form.